# A single-cell atlas of West African lungfish respiratory system reveals evolutionary adaptations to terrestrialization

Ruihua Zhang[1,2,3,13], Qun Liu[2,3,4,13], Shanshan Pan[2,3,13], Yingying Zhang[2,3], Yating Qin[2,3], Xiao Du[2,3,5], Zengbao Yuan[1,2,3], Yongrui Lu[6], Yue Song[2,3], Mengqi Zhang[2], Nannan Zhang[2,3], Jie Ma[2,3], Zhe Zhang[7], Xiaodong Jia[8], Kun Wang[9], Shunping He[6], Shanshan Liu[5,7], Ming Ni[5,7], Xin Liu [5], Xun Xu [5,10], Huanming Yang[5], Jian Wang[5], Inge Seim [11,12] ✉ & Guangyi Fan [2,3,5] ✉

The six species of lungfish possess both lungs and gills and are the closest extant relatives of tetrapods. Here, we report a single-cell transcriptome atlas of the West African lungfish (*Protopterus annectens*). This species manifests the most extreme form of terrestrialization, a life history strategy to survive dry periods that can last for years, characterized by dormancy and reversible adaptive changes of the gills and lungs. Our atlas highlights the cell type diversity of the West African lungfish, including gene expression consistent with phenotype changes of terrestrialization. Comparison with terrestrial tetrapods and ray-finned fishes reveals broad homology between the swim bladder and lung cell types as well as shared and idiosyncratic changes of the external gills of the West African lungfish and the internal gills of Atlantic salmon. The single-cell atlas presented here provides a valuable resource for further exploration of the respiratory system evolution in vertebrates and the diversity of lungfish terrestrialization.

The water-to-land transition of vertebrates—Terrestrialization—is a landmark event in evolutionary history that necessitated various adaptations to allow respiration in an aerial environment[1–5]. While gas exchange via gills facilitates respiration in water, a lung or swim bladder allows air breathing in a subset of fish species. Morphological and molecular data suggest the lung and swim bladder are homologous organs[2,5,6]. The first lung emerged in a common ancestor of ray-finned (Actinopterygii) and lobe-finned fishes (Sarcopterygii). The lung of ray-finned fishes was replaced (but retained by bichirs and the reedfish of the basal order Polypteriformes) by an air-breathing swim bladder still present in holostei fishes (the bowfin and gars) that was further repurposed by teleost fishes to facilitate buoyancy and, in some species, act as a sensory organ[2,5,7,8].

Lungfishes, a group of lobe-finned fishes, emerged around 400 million years ago and are the closest living relatives of tetrapods[9]. While it is appreciated that extant lungfish species have diverged in various ways since their Devonian origin[9,10] they nevertheless, as their name indicates, possess bimodal respiration with gills for water

[1]College of Life Sciences, University of Chinese Academy of Sciences, 100049 Beijing, China. [2]BGI Research, 266555 Qingdao, China. [3]Qingdao Key Laboratory of Marine Genomics, BGI Research, 266555 Qingdao, China. [4]Department of Biology, University of Copenhagen, Copenhagen 2100, Denmark. [5]BGI Research, 518083 Shenzhen, China. [6]State Key Laboratory of Freshwater Ecology and Biotechnology, Institute of Hydrobiology, Chinese Academy of Sciences, 430072 Wuhan, China. [7]MGI Tech, 518083 Shenzhen, China. [8]Joint Laboratory for Translational Medicine Research, Liaocheng People's Hospital, 252000 Liaocheng, Shandong, P.R. China. [9]Center for Ecological and Environmental Sciences, Northwestern Polytechnical University, 710072 Xi'an, China. [10]Guangdong Provincial Key Laboratory of Genome Read and Write, BGI Research, 518083 Shenzhen, China. [11]Integrative Biology Laboratory, College of Life Sciences, Nanjing Normal University, Nanjing, China. [12]School of Biology and Environmental Science, Queensland University of Technology, Brisbane 4000, Australia. [13]These authors contributed equally: Ruihua Zhang, Qun Liu, Shanshan Pan. ✉e-mail: inge@seimlab.org; fanguangyi@genomics.cn

breathing and lungs for air breathing. Understanding the respiratory characteristics of basal fish species with bimodal respiration promises to reveal insights into fundamental adaptations associated with early tetrapod terrestrialization and fish evolution[1–5].

There are six extant species of lungfish in three genera: the Australian lungfish (*Neoceratodus forsteri*) has a single lung, while the South American lungfish and four African species (*Protopterus* spp.) have two lungs[11,12]. African lungfishes and the South American lungfish are obligatory air-breathers—they depend on lung respiration and show limited oxygen uptake via their gills (i.e., their gills may be more important for carbon dioxide elimination, with species-specific differences observed)—while the Australian lungfish respires via its gills unless lung respiratory behavior is evoked naturally during short-term courtship behavior and flow events or experimentally by hypoxic water conditions[9].

A remarkable physiological characteristic of the African lungfishes is dormancy during the dry period that can last for months to years. To survive this hostile environment, they cease food and water intake and their blood pressure, heart rate, and respiration fall to low levels[9]. During this time, their gills are rendered non-functional for respiration, and these lungfishes rely exclusively on their lungs[13]. This process has been termed estivation by some investigators[14]. Others argue that 'estivation' is a misnomer since the Nigerian lungfish (*P. dolloi*) is an exception and does not downregulate its metabolism[9,15]. We will here employ the term terrestrialization coined by Perry and colleagues[15] to cover the continuum of terrestrial dormancy in lungfishes—with the West African lungfish (*P. annectens*) representing the most extreme form akin to true estivation. While the terrestrialized South African lungfish resides in a burrow, African lungfishes form a mucus cocoon to counteract water loss and pathogen invasions[16–18].

Molecular differences during African lungfish terrestrialization include immunologic adaptations associated with their cocoon and skin mucus[16,18], the NOS/NO system[13,19], and the expression of urea transporters[20], aquaporins[21], and hemoglobins[22]. Gene expression studies on lungfishes to date have considered bulk (homogenous) tissue from various organs. These include a combined transcriptome from various West African lungfish tissues[23], a skin transcriptome of aquatic and terrestrialized Nigerian lungfishes[16], and a combined heart and lung transcriptome of the East African lungfish[24]. While useful, de novo transcriptomes are a reduced representation of a genome and may not reliably or comprehensively capture the expressed gene set of a species if a genome is very large or complex[25]. Moreover, single-cell RNA-sequencing technologies now enable the full cellular diversity of organs to be characterized[26–29]. Two recent studies succeeded in sequencing, assembling, and annotating the >40-Gb genomes of the West African[4] and Australian[30] lungfishes. With two lungfish reference genomes in hand, it becomes possible to more systematically profile lungfish organs.

Here, we use single-cell RNA sequencing (scRNA-seq) to profile the transcriptional landscape of the lung and gill of the West African lungfish in aquatic and terrestrial environments. To validate cell-type-specific genes, we employ in situ hybridization (FISH). We also perform cross-species analysis to investigate the evolutionary conservation and heterogeneity of the West African lungfish lung and gill compared to fully terrestrial and aquatic species.

## Results

### A West African lungfish lung and gill cell type atlas
To study cell type diversity, we generated scRNA-seq data of lungs and gills from three West African lungfishes maintained under terrestrialized conditions for 33 days and three (Lung) or six (Gill) individuals from freshwater (Fig. 1a). We obtained 53,605 and 87,347 cells from 21 lung and 27 gill scRNA-seq libraries, respectively (Supplementary Fig. 1a, b) and detected more than 30,000 feature genes (Supplementary Data 1). The number of captured genes and cells is comparable to a bulk RNA-seq study of the East African lungfish lung[24] and published single-cell datasets of other aquatic species[31–34].

We applied Seurat[35] to identify marker genes and predict cell clusters and employed the unsupervised dimensionality reduction method UMAP[36] to visualize the clusters. We identified 14 lung cell types and 22 gill cell types (Fig. 1b–e; Supplementary Fig. 1c, d) (Supplementary Data 2). We also compared the lung and gill cell type repertoires, revealing broad cell type similarities as well as organ-specific cell types (e.g., alveolar cells in the lung) (Supplementary Fig. 1e). We next focused on cell type annotation and their marker genes.

In the lung, our analysis revealed epithelial cells, stromal cells, endothelial cells, and immune cells; with a large number of alveolar cells (selected marker genes are shown in Supplementary Fig. 1c). Our cell type classification is consistent with an electron microscope study of the marbled lungfish (*P. aethiopicus*) showing obvious alveolar epithelium cells, elastic tissue, stromal cells, and blood vessels[37]. We identified alveolar epithelium cells in our data using the gene *slc34a2*, which has a phosphate transport function and is highly expressed in the apical membrane of human type II alveolar epithelium cells (ATII cells)[38,39]. Interestingly, morphometric and single-cell transcriptome appraisals of mammalian lungs suggest that they have a much smaller proportion of alveolar epithelial cells (about 20%) but additional alveolar and immune cell subtypes (about 40–60%)[28,40,41]. Terrestrial mammals harbor various alveolar cell types with canonical marker genes[42,43], while expression patterns in our scRNA-seq data and electron-microscopy of the marbled lungfish[37] did not effectively stratify further alveolar epithelium cell subtypes. Similarly, a recent cell atlas of *Xenopus* lung identified a single alveolar epithelium cell type[44]. The stromal marker decorin (*dcn*)[45], an extracellular matrix protein, was used to classify stromal cells. The gene *arpc1b*, which plays a role in inflammatory diseases[46] and has macrophage-specific expression in human GTEx data analyzed by the Human Protein Atlas[8], showed high expression levels in lung macrophages. We classified other cell clusters using general markers of erythroid cells (e.g., *hbb* and *hba2*), vascular endothelial cells (e.g., *egfl7* and *vwf*), epithelial cells other than alveolar epithelium (e.g., *krt8*, *krt17* and *gsta1*), and lymphoid cells (e.g., *sh2d1a*).

We also identified various cell types in the gills of the West African lungfish (selected marker genes are shown in Supplementary Fig. 1d). The number of immune cells in the gill (under freshwater and terrestrialization conditions) is more than that of the lungs, likely because that its external gills are in direct contact with water and continuously exposed to toxins and pathogens. We identified major lymphoid cell types in the West African lungfish gill including T cell, CD8+ T cell, B cell and NK-like cell (Supplementary Fig. 1d). We found the NK-like cell cluster expressed representative markers of mammal granzyme M (*GZMM*) and granzyme A (*GZMA*)[47,48], as well as genes denoted NK cell type-specific in the molecular signatures database (MSigDB) (Supplementary Data 3). Rare cell types identified included platelet cells, which show high expression of *TIMP4* (the major inhibitors of matrix metalloproteinases in human platelets) and *F13A1* (coagulation factor XIII, A1 polypeptide). We identified two types of ionocytes (ionocyte cells and ionocyte_dmrt2 cells), cells that regulate salt balance and show environment-induced gene expression plasticity of ion transporters[49], agreeing with West African lungfish scanning-electron[50] and immunofluorescence micrographs[51].

To validate the cell types classified using marker genes, we performed H&E staining and RNA fluorescence in situ hybridization (FISH) on cross-sections of West African lungfish lungs and gills. The FISH results were consistent with the physiological structure of the marbled lungfish lung, the mammalian lung, and gill filaments of ray-finned fishes (Fig. 1f, g; Supplementary Fig. 1f)[37,52–55]. Moreover, Gene Set Enrichment Analysis (GSEA)[56,57] of genes differentially expressed by cell types revealed largely cell-type specific enrichment, consistent

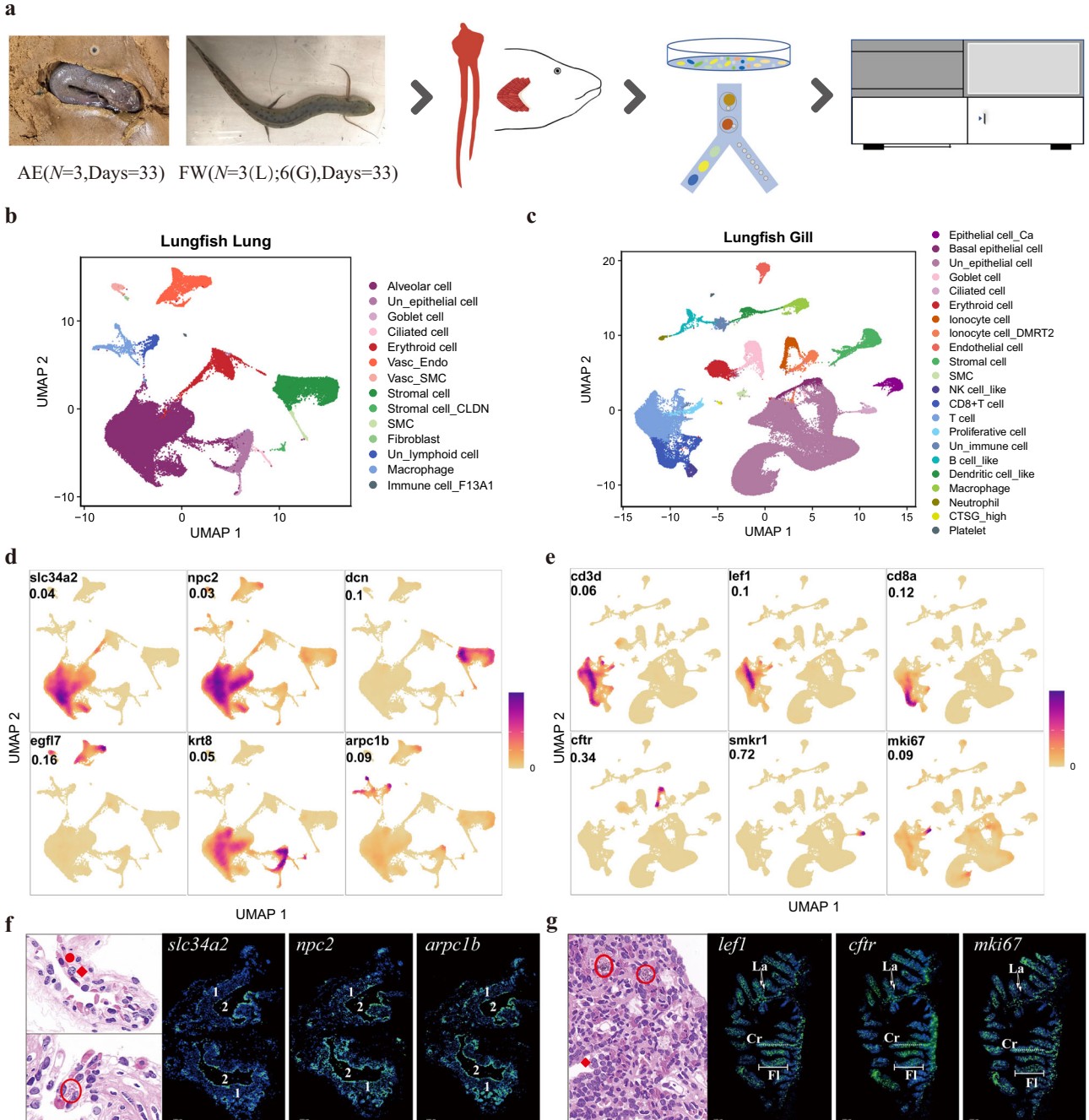

**Fig. 1 | Single-cell transcriptional profiles of West African lungfish lung and gill under freshwater and terrestrialized conditions. a** Experimental workflow in this study. AE denotes terrestrialization; FW, freshwater. L, lung; G, gill. **b** Lung cell landscape of 53,605 cells from terrestrialized and freshwater lungfish. Cells are shown by Uniform Manifold Approximation and Projection and color-coded by cluster cell type. Each dot represents a cell, and different colors are associated with specific cell types. Vasc_Endo, vascular endothelial cell; Vasc_SMC, vascular smooth muscle cell; SMC, smooth muscle cell; Un_epithelial cell, unclassified epithelial cell; Un_lymphoid cell, unclassified lymphoid cell. **c** Gill cell landscape of 87,347 cells from terrestrialized and freshwater lungfish. Cells are shown by UMAP and color-coded by cluster cell type. Each dot represents a cell, and different colors are associated with specific cell types. NK cell_like, Natural killer cell_like; Un_epithelial cell, unclassified epithelial cell; Un_immune cell, unclassified immune cell. **d** Cell type identification of six classical lung cell type markers (see Supplementary Fig. 1c

for overall markers). Markers are shown by density plots. Values indicate the max density. The color scale is relative, ranging from 0 to highest density. **e** Cell type identification of six classical gill cell type markers (see Supplementary Fig. 1d for overall markers). Annotated as in (**d**). **f** Confirmation of scRNA-Seq cell type annotation results by H&E stanning and fluorescence microscopy image of African lungfish lung. Green, digoxigenin-labeled marker genes probes amplified using FITC-TSA; blue, DAPI (scale bar = 500 μm). (1) fibromuscular wall; (2), air sacs. On the H&E image, the red solid circle represents alveolar epithelial cells; red rhombus represents macrophage cells; red circle represents the lymphoid nodes. Each slide was repeated independently 3–5 times. Scalebar, 500μm. **g** Confirmation of scRNA-Seq cell type annotation results by H&E stanning and fluorescence microscopy image of African lungfish gill. Annotated as in (**f**). Fl, filament; Cr, cartilage; Wight arrows, lamella. Each slide was repeated independently 3–5 times.

with cell type classifications (Supplementary Fig. 2a, b; Supplementary Data 3).

## Cellular and molecular features of lungfish lungs during terrestrialization

Lungfish terrestrialization, observed in the South American lungfish and African lungfishes, is an episodic response to the extreme environment during dry periods and are reversible adaptive changes of the gills and lungs[9]. A period of dormancy ('the maintenance phase'), which can last for years, then follows. While the West African lungfish is an obligate air-breather, its respiratory organs nevertheless show fundamental morphological changes during the maintenance phase: the paired lungs become expanded and better vascularized (e.g., see refs. 13,17).

We performed scRNA-seq after 33 days at the maintenance stage, where the West African lungfish exhibit a lowered oxygen consumption and respiratory rate, and compared its expression to fish maintained in freshwater. We focus primarily on the lung since its comparatively similar structure during freshwater conditions should allow for more biologically relevant gene expression comparisons. Few discernible differences in lung and gill cell types were detected after 33 days at the maintenance stage compared to freshwater lungfish (Supplementary Fig. 3a). To further characterize molecular changes in cell types during West African lungfish terrestrialization, we performed differential gene expression analysis and observed more downregulated DEGs in 14 cell types of the lung and 16 gill cell types (Supplementary Data 4). In agreement with a lowered respiratory rate during terrestrialization, erythroid cells (red blood cells) showed reduced expression of the aminolevulinate synthase isoform genes *alas1* and *alas2*, enzymes that catalyzes the heme biosynthetic pathway, in both the lung and gill after 33 days at the maintenance stage (Supplementary Data 5, 6).

In the lung, the alveolar epithelium plays a fundamental and essential role in gas exchange. In this cell type, we identified 328 upregulated and 639 downregulated DEGs (Supplementary Data 5). Genes encoding the electron transport chain essential for cellular energy production were downregulated[58] (Fig. 2a, b), likely reflecting a reduced metabolic rate despite a larger alveolar epithelium cell number. Other downregulated genes of interest include the solute carrier proteins slc34a2 and slc22a3 (Fig. 2c). Lung alveolar epithelial genes upregulated in terrestrialized lungfish included *npc2* (NPC intracellular cholesterol transporter 2), which has a role in lung lipid metabolism and is associated with Niemann–Pick disease characterized by alveolar proteinosis[59], and the antioxidant enzyme genes *mgst1* (microsomal glutathione *S*-transferase 1) and *prdx1* (peroxiredoxin-1) (Fig. 2c). Increased alveolar epithelium expression of antioxidant genes expands on a previous biochemical study showing an enhanced antioxidant capacity of the Nigerian lungfish heart and brain during terrestrialization[60]. Genes such as *dnajb13*, *ccdc39* and *dnaaf1* (Fig. 2c) were upregulated in ciliated cells and likely reflect the expanded lung under terrestrialization. Genes upregulated by lung macrophages during terrestrialization were enriched for the endocytosis and phagocytosis pathways (Fig. 2d). Genes of interest include *apoe* (apolipoprotein e) and *marco* (macrophage receptor with collagenous structure). Apoe is secreted from macrophages and plays an important role in immunoregulation[61]. Marco is a scavenger receptor that plays a crucial role in the immune response against microbial infections[62]. The vascularized lungfish lungs downregulated *vwf* (von Willebrand factor) and *selp* (P-selectin) in endothelial cells, which may reflect a reduced blood clotting ability to prevent intravascular thrombosis[63], as previously reported for the terrestrialized West African lungfish liver[64].

We next considered cellular crosstalk between different cell types in freshwater and during terrestrialization using CellPhoneDB, a database of ligand and receptor interactions that considers their subunit architecture[65]. We detected 35 significant ligand-receptor pairs (LR pairs) among the 14 lung cell types (Fig. 2e; Supplementary Fig. 3b, c). The freshwater lungfish showed more connections among different cell types in the lung (31 LR pairs; 15 shared interactions with terrestrialized lungfish) (Supplementary Fig. 3d). To better understand the cell type interactions, we performed GO enrichment analysis on ligand-receptor pair genes from the lung, revealing enrichment for blood vessel formation (Fig. 2f).

We considered LR pairs with differential gene expression during terrestrialization and identified two pairs with changes in the lung: nrp1:vegfa[66,67] and nrp1:sema3a[68] (Fig. 2g). The transmembrane glycoprotein neuropilin-1 (nrp1) is a coreceptor of vascular endothelial growth factor A (vegfa) and semaphorin-3A (sema3a)[69]. The binding of NRP1 on the surface of endothelial cells by VEGFA stimulates vascularization (i.e., blood vessel formation) via vasculogenesis and angiogenesis, while the combination of NRP1 and SEMA3A inhibits vasculogenesis[70,71] (Fig. 2g). VEGFA and NPR1 show lung-specific expression in species ranging from basal ray-finned fishes (e.g., bichirs), to lobe-finned fishes (lungfishes and coelacanths) to tetrapods[5]. Interestingly, sarcopterygians (lungfish, coelacanth, and tetrapod) *nrp1* loci share a candidate enhancer speculated to facilitate the vertebrate water-to-land transition[4]. While further studies are needed to validate signaling interactions between West African lungfish cell types (in particular the contribution of non-epithelial cell ligands to vascularization mediated by endothelial cells), we observed that the freshwater lungs showed higher expression of *nrp1:vegfa* LR pairs (Fig. 2h). This included higher expression of *npr1* in endothelial cells and alveolar cells of the lung and higher expression of *vegfa* in alveolar cells of the lung (Supplementary Fig. 3e). The expression level of *sema3a* was too low to conclude an antagonist role in the West African lungfish. However, a previous study found that the lung of terrestrialized West African lungfish are better vascularized and expanded[13], hinting that SEMA3A outcompetes VEGFA vascularization effects under freshwater conditions despite its high gene expression.

## Difference in gills between freshwater and terrestrialized condition

African lungfishes are near-obligate air-breathers in freshwater and acquire ~90% of their oxygen via their lungs, while ~10% is acquired via the gills and skin[13,17,72]. The manifestation of terrestrialization in the two respiratory organs are vastly different. After about a week of terrestrialization, the West African lungfish forms an immunoprotective cocoon from mucus secreted by its gills (this study and[16–18,64,73]). Thus its gills are rendered non-functional and covered in mucus during terrestrialization.

Nevertheless, most of the carbon dioxide is released via ionocytes of the fish gill epithelium in freshwater[74]. As such, the African lungfish gill may be better thought of as a carbon dioxide-release organ. During terrestrialization, lungfish gills regress and become covered in mucus, and the lung takes over the carbon dioxide release role.

Mitochondria-rich ionocyte cells play a dominant role in osmotic and acid-base regulation[75–77]. After 33 days at the maintenance phase (Supplementary Fig. 4a), we identified 1,552 and 1,130 downregulated DEGs in 'ionocyte' and 'ionocyte_dmrt2' cells, respectively (Supplementary Data 6). Of these DEGs, 867 were shared by the two cell types and showed enrichment for mitochondrial localization (Supplementary Fig. 4b). The 685/1,552 DEGs uniquely downregulated in 'ionocyte' cells were enriched for biological processes associated with ion transport (Supplementary Fig. 4c), in agreement with the gill osmoregulatatory function[78]. Given the various types of immune cells in the West African lungfish gill, we grouped them, using the human classification system[79], into myeloid (dendritic cells and macrophages) and lymphoid (i.e., T cell, NK cell, and other lymphoid cell) lineages. Myeloid cells are part of the innate immune system[80–82], while lymphoid cells form part of the adaptive immune response[83,84].

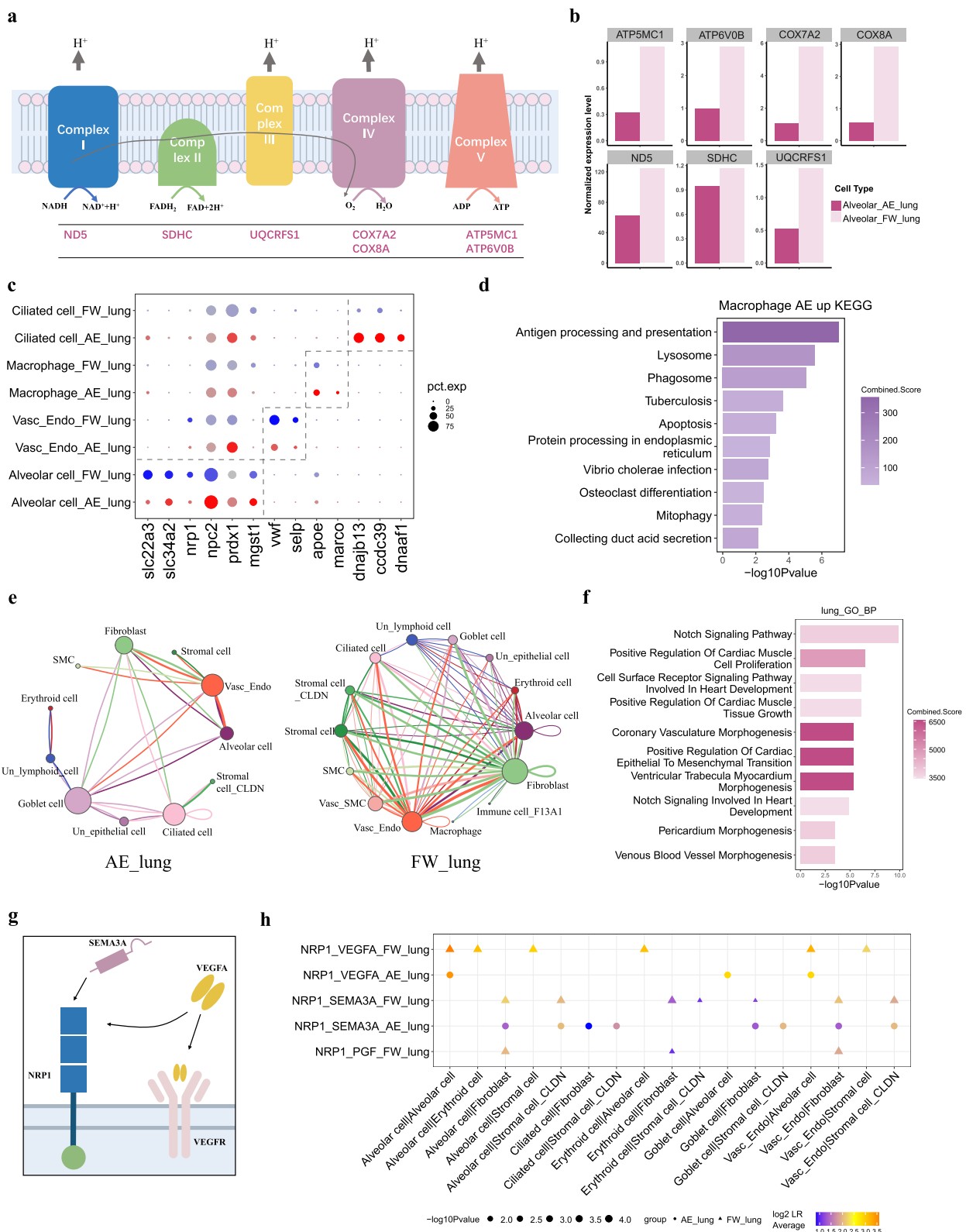

We observed relatively few myeloid cells, however, these cells are shorter-lived than most lymphoid cells[85] and our single sampling point (33 days) may not have adequately captured their cell dynamics at the maintenance phase. We observed that many glycoproteins, key to both the innate and adaptive immune systems[86] (e.g., *cd226*, *cd59*, and *cd5*) and regulatory factors (e.g., *lck* and *lcp1*), showed lower expression in most immune cells of the gill at the maintenance phase (Fig. 3a).

However, genes such as *cxcl2*, *pmp22* and *abcc9* were highly expressed in myeloid cells of terrestrialized gills, accordance with the roles of granulocyte in cocoon under terrestrialization[16] (Fig. 3a). In proliferative cells (ostensibly a proliferative lymphoid cell subset such as T cells, see UMAP clustering in Fig. 1c), we observed a downregulation of genes encoding antioxidant enzymes and genes associated with mitochondrial bioenergetics (Fig. 3b, c and Supplementary Data 6).

**Fig. 2 | Cellular and molecular features of lungfish lungs during terrestrialization. a** Schematics of the electron transport chain with differentially expressed genes (DEGs) in lung alveolar epithelial cells indicated. **b** The average expression levels of DEGs (see **a**) in alveolar cells during terrestrialization (AE; in maroon) and in freshwater (FW; in salmon) by bar plot. **c** Comparisons of selected DEGs between freshwater and terrestrialized lung cell types by bubble plot. AE denotes terrestrialization (red bubbles); FW, freshwater (blue bubbles). Circle size reflects the percentage of cells within a cell type which express the specific genes. **d** Enriched KEGG pathway of AE macrophage up regulated DEGs. The enrichment analysis was generated via the Enrichr web server using Fisher exact test, and the Benjamini-Hochberg (BH) was used for correction for multiple hypotheses testing. *P* values are indicated on the *x*-axis. **e** Comparison of cellular interactions among all cell types in

terrestrialized lungfish lung (AE_lung, left) and freshwater lungfish lung (FW_lung, right) by net plots. Each node represents a kind of cell type, links mean the legend-receptor interactions and the width of link is in direct proportion to the LR pair numbers. **f** Enriched biological processes of LR pairs in the lungfish lung. Generated using the Enrichr web server. Annotated as in Fig. 2d. **g** Schematics of the trans-membrane coreceptor NRP1 and its ligands VEGF and SEMA3A. **h** The expression of selected LR pairs in lung cell types from freshwater (FW) and terrestrialized (AE) lungfish. Circle (AE) or triangle (FW) size reflects the percentage of cells within a cell type which express the specific genes. The color of circles/triangles indicates the average expression of the LR pair. *P* values were calculated by CellphoneDB, which means the likelihood of cell-type specificity of a given receptor–ligand pairs. Source data are provided as a Source Data file.

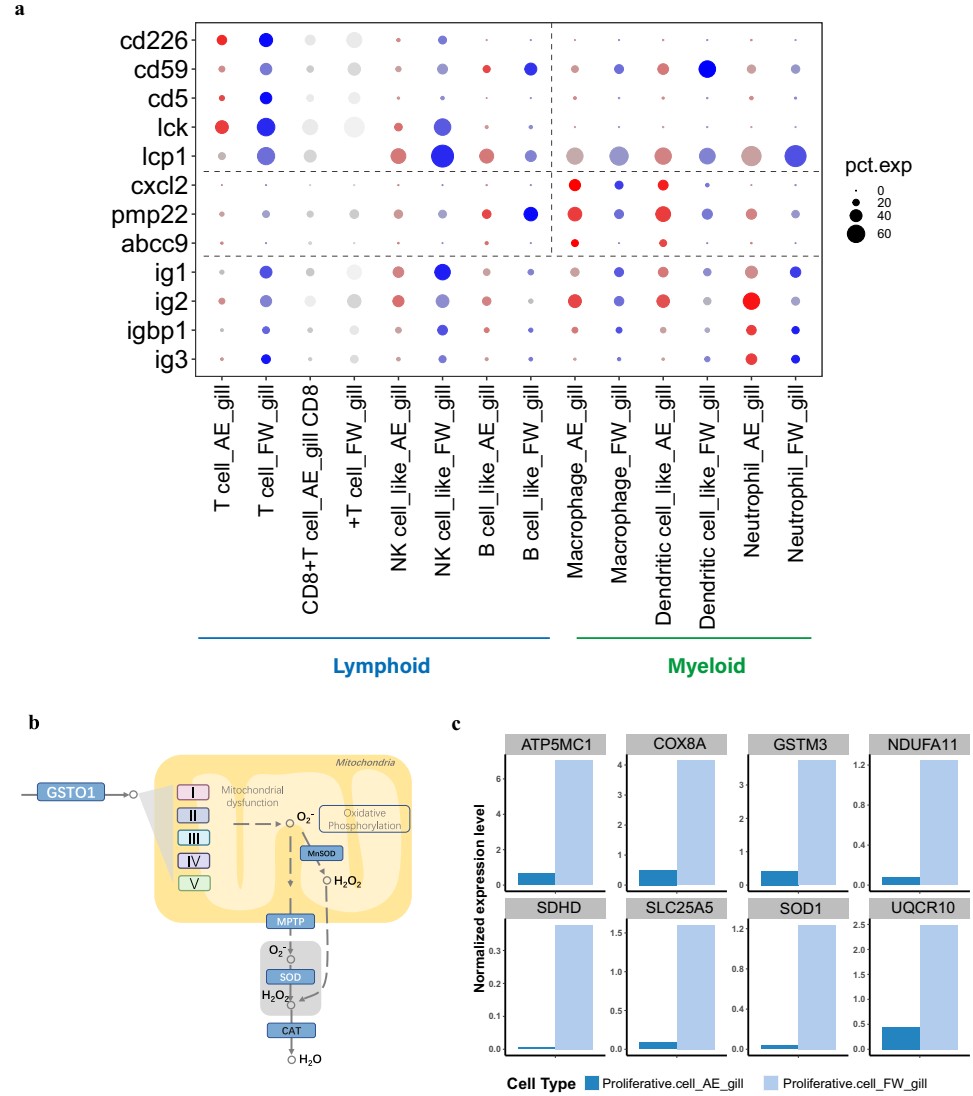

**Fig. 3 | Difference in gills between freshwater and terrestrialized condition. a** Comparison of selected DEGs between freshwater and terrestrialized gill cell types by bubble plot. Annotated as in Fig.2c. **b** Schematics of the reactive oxygen species pathway differentially expressed genes (DEGs) in gill proliferative cells indicated. **c** The average expression levels of DEGs (see Fig.2b) in gill proliferative cells during terrestrialization (AE; in dark blue) and in freshwater (FW; in light blue).

We also considered cell-cell communications in lungfish gill data. We detected 51 significant ligand-receptor pairs (LR pairs) across 22 gill cell types (Fig. 4a) and the freshwater lungfish gill data showed more connections than terrestrialized (Supplementary Fig. 4d). GO analysis of total gill ligand-receptor pair genes showed enrichment for vasculature morphogenesis (Fig. 4b). In contrast to the lung data, the

*nrp1:vegfa* LR pair showed higher expression in terrestrialization gills (Fig. 4c). This included higher expression of *npr1* in endothelial cells and higher expression of *vegfa* in stromal cells of the gill (Supplementary Fig. 4e). This is consistent with the observation that the vasculature of marbled lungfish is dilated but not collapsed during terrestrialization[87].

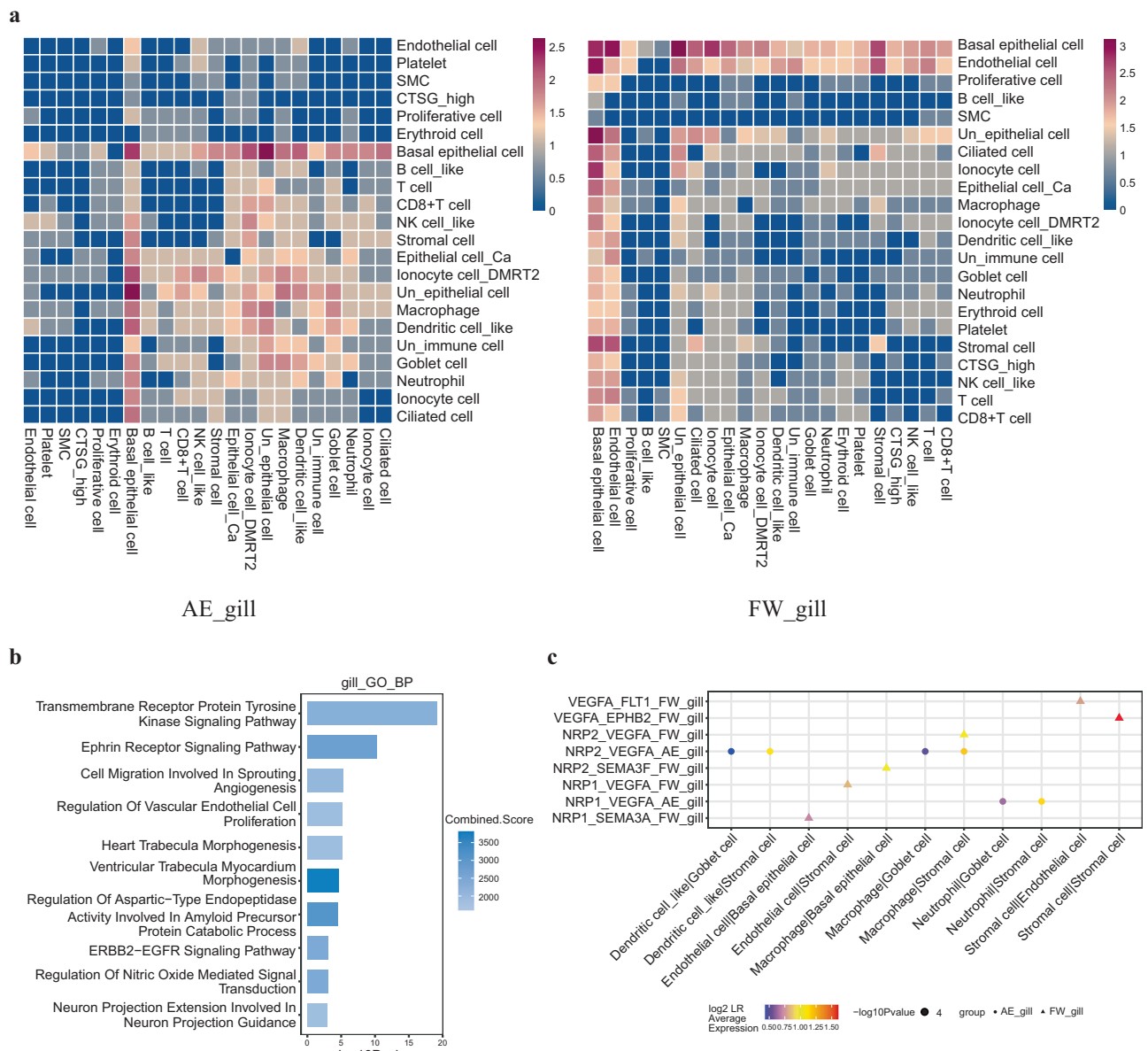

**Fig. 4 | Cell-cell communication of gills between freshwater and terrestrialized condition. a** Comparison of cell-cell communications between cell types in terrestrialized lungfish gill (left) and freshwater lungfish gill (right) by heatmap. Color means the log-normalized numbers of CellPhoneDB ligand-receptor pairs (LR pairs) **b** Enriched biological processes of LR pairs in the lungfish gill. The enrichment analysis was generated via the Enrichr web server using Fisher exact test, and the Benjamini-Hochberg (BH) was used for correction for multiple hypotheses testing. *P* values are indicated on the *x*-axis. **c** The expression of selected LR pairs in gill cell types of freshwater (FW) and terrestrialized (AE) lungfish. Annotated as in Fig. 2h. Source data are provided as a Source Data file.

## Cell type evolution of the lung and gill

Genetic and fossil evidence suggests that lungfishes are pivotal species for studying the vertebrate water-to-land transition[4,88,89]. We integrated our West African lungfish scRNA dataset from freshwater with atlases of the human lung[90], mouse lung[90], zebrafish swim bladder[91], Atlantic salmon gill[92], and zebrafish gill[91] (Supplementary Fig. 5a–g; Supplementary Fig. 6a, b and Supplementary Data 7).

We found that epithelial cells and granulocyte cells of the lung and swim bladder show high similarity among these four species, suggesting that these cell types share ancestral cell types (Fig. 5a; Supplementary Fig. 5d–f; Supplementary Fig. 6b). As for the comparison between lungfish and mammals, alveolar epithelium cells, ciliated cells, and lymphoid cells of West African lungfish lung showed high similarity to mammalian (here: human and mouse) ATII cells, ciliated cells, and T cells. However, many immune cell types in the human or mouse lung show no correlations with the West African lungfish

(Supplementary Fig. 5e, f), likely reflecting the distinct immune cell repertoire of these terrestrial mammals. In summary, our data indicate broad homology between the swim bladder of ray-finned fishes (here, the zebrafish) and the lungs of tetrapods and the West African lungfish. Interestingly, the cell type correlations between the zebrafish swim bladder and West African lungfish lung are mainly related to innate immune function and the structural characteristic of gas-filled organs. In contrast, the correlations to human lung cell types are related to air breathing and adaptive immunity.

Because the comparison of the various cell type atlases revealed broadly similar cell types, we further explored the gene sets underlying the respiratory organ cell types of the West African lungfish and other species. We found that 49 genes with a ZFIN (Zebrafish Information Network)[93] swim bladder development phenotype (Supplementary Data 8) also show cell type-specific expression in the West African lungfish lung (Supplementary Fig. 5h). Particularly genes expressed by

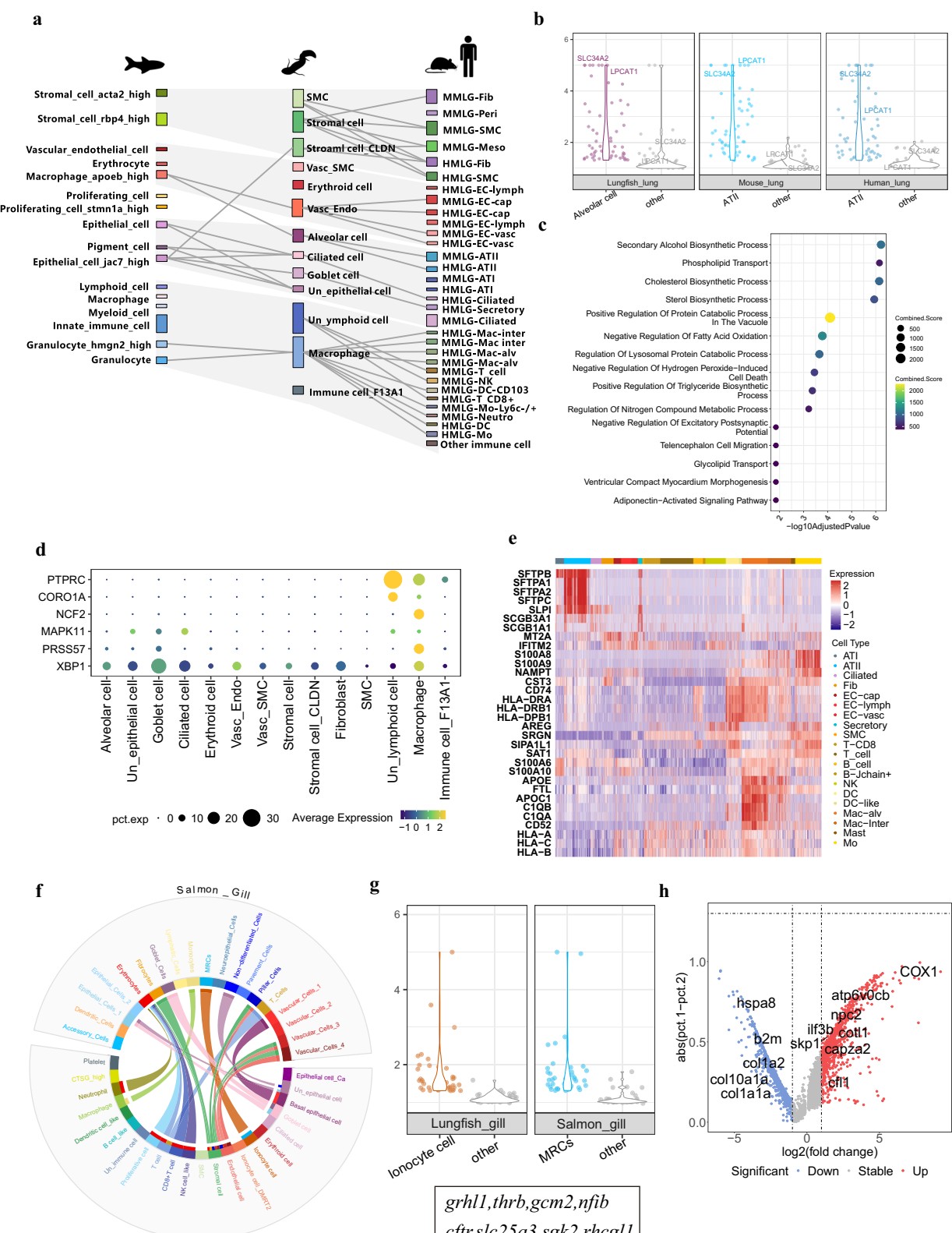

the stromal cells and epithelial cells—including *srm* (spermidine synthase)[94], *wwtr1* (WW domain containing transcription regulator 1)[95], *hdac1* (histone deacetylase 1)[96], *cfl1* (cofilin 1)[97], and *clu* (clusterin)[98]—result in abnormal phenotype of swim bladder formation.

In addition, the West African lungfish, mouse, and human alveolar epithelium cell type is characterized by 57 orthologs, including three transcription factors (*gata6*, *irx2*, and *mecom*) (Fig. 5b), suggesting that

these genes have mediated core functions in this cell type since the emergence of the first lung ~400 million years ago[4]. The genes *slc34a2* and *lpcat1* are expressed highly in alveolar cells across species. *Slc34a2* is mainly expressed in ATII cells of human lung and plays a role in regulating phosphate levels, while *lpcat1* is essential for air breathing by mammals[99–102]. The transcription factors *gata6* and *irx2* involved in mouse lung development also showed high expression in lungfish lung

**Fig. 5 | Cell type evolution of the lung and gill. a** Pairwise cell type similarities of the zebrafish swim bladder, lungfish lung, mouse lung, and human lung by sankey plot. The top 5% highest Kullback–Leibler divergence (KDL) values are indicated as arches connecting cell types for each pair. **b** Conserved expression in alveolar cells of the human lung, lungfish lung, and mouse lung. Violin plots are including 57 orthologous genes (>1.3 log2 fold-change). Each dot represents a gene, and normalized average expression values are shown. **c** Enriched biological processes of 57 orthologous genes in b. The enrichment analysis was generated via the Enrichr web server using Fisher exact test, and the Benjamini-Hochberg (BH) was used for correction for multiple hypotheses testing. *P* values are indicated on the *x*-axis. **d** The expression of selected DEGs in lungfish-zebrafish-human-mouse comparisons in the lungfish lung. Circle size reflects the percentage of cells within a cell type that expresses the specific genes. The color of circles indicates the average expression of a DEG. **e** The expression of lung cell type DEGs in lungfish compared to human. Mean values are shown. DEGs with a *q*-value ≤ 0.05 and fold change ≥3 are shown. **f** Cell–cell similarities between the gills of the lungfish and Atlantic salmon. The top 5% highest Kullback–Leibler divergence (KDL) values are indicated as arches connecting cell types. **g** Violin plots of 40 orthologous genes (>1.3 log2 fold-change) with conserved expression in MRCs of the salmon gills, lungfish gills.Annotated as in (**b**). **h** The expression of DEGs of lungfish-Atlantic salmon gill comparisons in the lungfish gill. The *x*-axis of volcano plot shows the log2 fold-change, the *y*-axis represents the percent of lungfish and salmon gill cells that express each gene (calculating the Δ percentage difference). Source data are provided as a Source Data file.

(Supplementary Data 9). Furthermore, the 57 orthologs showed enrichment for phosphate symporter activity and phosphatidylcholine flippase activity (Fig. 5c), suggesting a fundamental function of phosphate metabolism across species. Macrophages of the lung are a major cell type for defense. We found that the zebrafish, lungfish, mouse, and human express a shared gene set with high expression level in granulocyte and other immune cells (Fig. 5d; Supplementary Fig. 5i–k).

We also noted cell type diversification in the lung atlas of the African lungfish and humans (Supplementary Data 10). The downregulated DEGs included the pulmonary surfactant proteins (*SFTPB*, *SFTPA1*, *SFTPA2*, and *SFTPC*) that play a role in maintaining alveolar structural stability and immune response[103] (Fig. 5e). Many other genes with immune cell type expression and function (Fig. 5e), such as calcium-binding protein S100A8 and S100A9[104], apolipoprotein APOE and APOC1[105], cell surface protein CD74[106], were also downregulated DEGs in the West African lungfish.

We next considered cell type conservation and diversification of the West African gill by comparing its freshwater atlas with the zebrafish and Atlantic salmon. We found that ionocyte cells, vascular cells, lymphoid cells, T cells, and macrophages exhibited high similarity across these species (Fig. 5f; Supplementary Fig. 5g; Supplementary Fig. 6a), suggesting that the greatly reduced gills of the West African lungfish[9] retains major functions such as osmoregulation, material transportation, and immunity. Gill ionocyte cells play a dominant role in osmotic and acid-base regulation in fish metabolism. We found the ionocyte cells in lungfish gills shared 40 orthologs including four transcription factors (*grhl1*, *thrb*, *gcm2*, and *nfib*) (Supplementary Data 9). Among them, *grhl1* and *nfib* play a role in development, and *cftr*, *slc25a3*, *sgk2*, and *rhcgl1* play a role in ion transport of chloridion, phosphate, potassium, and ammonia (Fig. 5g). A comparison between all cell types in the gill atlases of the West African lungfish and Atlantic salmon, revealed 5253 DEGs (3,486 upregulated and 1,767 downregulated) in the West African lungfish (Fig. 5h; Supplementary Data 11). The genes upregulated included immune genes such as *ilf3* and *skp1* of the innate immune system[107,108] and genes associated with actin dynamics (*cfl1*, *cotl1*, and *capza2*). Genes downregulated in the gill included the collagens *col1a1a*, *col1a2*, and *col10a1a* (Fig. 5h). Taken together, genes downregulated in the West African lungfish gill likely play a role in the basic structure and function of gill, hinting at its relatively simple structure compared the gills of ray-finned fishes. The immune genes upregulated may reflect that it has external gills that are exposed to the environment. In contrast, the internal gills of the Atlantic salmon are separated from the environment by its gill slits[109].

## Discussion

In this study, we employed single-cell RNA-sequencing to construct a cell-type transcriptome atlas of the West African lungfish lungs and gills. The cell type atlas showed a cellular diversity of respiratory organs in agreement with the literature, including alveolar epithelium cells of the lung and ionocyte cells of the gill epithelium[37,50]. While previous studies comparing lung and gill cell types of African lungfishes were exclusively morphological[37,87,110–113], our single-cell transcriptome lens allowed a finer picture. We identified a single alveolar epithelium cell type, supporting morphological data[37]. Together with an amphibian lung scRNA-seq study[44], the data support that two distinct alveolar epithelium cell types are an amniote innovation[28,90]. It is important to note that current approaches to characterize cell types rely heavily on published markers from model animals. That is, particularly West African lung fish cell types can be hard to classify or even identify (e.g., CTSG_high cells in the gills with high levels of the antimicrobial protein cathepsin G in Fig. 1c).

Our dataset revealed that establishing the immune cell repertoire of the West African lungfish requires further studies. Two factors contribute to this interpretation. First, immune cells in the circulation may overestimate immune cell content. Biological replicate experiments where we introduced additional washing steps to remove blood revealed a pronounced reduction in immune cell number in the gill but not the lung. Secondly, lung immune cells in our dataset are likely solely intraepithelial in origin because the lung samples were prepared using *Bacillus licheniformis* protease (see[114]), an enzyme that does not dissociate immune cells from the lamina propria. Given these limitations, future studies should further optimize methods for isolating lungfish respiratory compartments (e.g., see refs. [115,116]). Despite possibly not capturing the full gill and lung immune cell repertoire, we profiled the organs cell type landscape.

We employed our cell-type atlas to profile the lung and gill of the West African lungfish during terrestrialization (at the maintenance stage, 33 days after cocoon formation in dry mud). This effort revealed gene expression in broad agreement with the reversible morphological and physiological changes reported in the literature[9]—in particular, the remarkable regression (and ceased $CO_2$ gas exchange) of gills coupled with expansions and vascularization of the lungs to facilitate exclusive $O_2$ and $CO_2$ gas exchange. These results provide an impetus for generating scRNA transcriptome atlases from the remaining four terrestrializing lungfish species (three *Protopterus* spp. and *Lepidosiren paradoxa*), as well as the non-terrestrializing Australian lungfish (*Neoceratodus forsteri*), to cover the continuum of terrestrial dormancy in lungfishes.

Evidence is emerging that cell types are more suitable 'evolutionary units' than tissues or organs[29,44,91]. Single-cell transcriptome atlases of many vertebrates are now available[28,44,90–92]. We performed cross-species integration to compare the cellular repertoires of fully terrestrial mammal lungs (human and mouse), the gills of teleost ray-finned fishes (zebrafish and Atlantic salmon), and the swim bladder of the zebrafish to the West African lungfish. Some basal freshwater ray-finned fish species have a respiratory swim bladder for air-breathing (e.g., the bowfin and gars), while the swim bladder of most fishes (e.g., zebrafish examined here) has been repurposed to strictly facilitate buoyancy or act as a sensory organ[7]. Our cell type-level transcriptome analysis suggests that: (1) despite more than 400 million years of evolution, there is broad homology between the swim bladder of ray-finned fishes and lungs of fully terrestrial tetrapods and the West African lungfish—in agreement with the literature[2,5,6]; and (2) although regressed compared to the internal gills of ray-finned fishes, the

external lungfish gills retain major functions such as ion regulation and gas exchange.

In conclusion, we have constructed the first single-cell transcriptome atlas of a lungfish, that of the West African lungfish (*Protopterus annectens*). We show that the atlas is suitable for the juxtaposition of the respiratory organs of lungfish and cross-species analysis. The atlas provides a rich resource for understanding the evolution and adaptations of the vertebrate respiratory system.

## Methods

### Ethics and experimental animals
This study was performed in accordance with the guideline of the national and organizational stipulation. All the process of animal transportation, feeding, the terrestrialization experiment and all aspects of the animal experiments (including animal killing and dissection) were approved by the Institutional Review Board on Ethics Committee of BGI (NO. FT19057-T1). All the West African lungfish (*Protopterus annectens*) samples (weight 67–106 g, body length ~30 cm) were purchased from an aquarium market in Guangzhou, China, and kept in oxygen-rich water during transportation to our laboratory. All the specimens were about 2 years old. There were 11 experiment animals used in our study, including 7 animals in freshwater group and 4 animals in terrestrialization group. In terrestrialization group, three animals were sampled lung and gill tissues to perform scRNA-seq, and lung and gill from another animal were used to perform H&E stain and in situ hybridization. In freshwater group, six animals were sampled lung and gill tissues to perform scRNA-seq, of which three lung tissues were used to perform preliminary experiment, and the other three lung tissues and six gill tissues were used in the formal experiment and sequencing. One animal in freshwater group was used to perform H&E stain and in situ hybridization. The detailed information on libraries and animals was in Supplementary Data 1. We did not sex the specimens.

### Terrestrialization experiment
All West African lungfish specimens were kept in three plastic tanks (W25 cm × L30 cm × H35 cm) with aerated tap water for a week to fit the lab environment. The water was kept at 20 cm, and half was changed daily. For the African lungfish terrestrialization experiment, we prepared two additional plastic tanks and dried mud to build a terrestrialization environment. Fine, dried mud was purchased from a garden shop. We used adequate aerated tap water to soak the dried mud (20 kg × 3 tanks) for two days to keep it completely wet[73]. The height of the mud should be at least 15 cm. The West African lungfish were both kept in each tank for 33 days. We sprayed a little water (about 20 ml) on the surface when the mud cracked. All the specimens were not provided with food after the commencement of the experiment.

### Tissue collection and cell preparation
On day 34, we prepared the terrestrialized and freshwater samples and performed the subsequent process. For the terrestrialized lungfish, we dug it out of the mud and rinsed off any remaining mud with water. All specimens were killed by a blow to the head. Dissection and tissue collection were finished within 30 min and performed on ice. Briefly, the ventral body wall was opened and the liver, gall bladder, and digestive tract were separated and removed. Another scalpel splits the mucous membrane tissue from the spine and kidneys. Clamped paired lungs were clipped nearly to the throat and immediately put into the ice-cold PBS solution in a culture dish. The paired gills were also clamped and lifted by tweezers, the junction of the branchial arch was cut off using a scalpel, and the gills were put into ice-cold PBS. All tissues in PBS were visually inspected, and visible blood clots and muscular tissue were removed by using a scalpel.

After the tissue collection, tissue disaggregation and enzymatic dissociation were performed independently. Following dissociation, the resulting suspension was passed through a 40 μm filter (BD # 352340), and the filter was washed with 2 ml cold PBS to ensure efficient digestion and sufficient collection. The filter suspension was transferred to a 15 ml centrifuge tube and centrifuged at 4 °C with 300$g$ for 5 min. Then the supernatant was removed, and the centrifugation sediment was washed with 5 ml PBS + 0.04%BSA and centrifuged again (4 °C at 300$g$ for 5 min), and the centrifugation sediment was rewashed with 5 ml PBS + 0.04%BSA. After the centrifugation, the cell pellets were re-suspended in 300 μl PBS + 0.04% BSA. All samples were stained, and cell viability was measured using Trypan Blue (Invitrogen) and manually counted using a hemocytometer (INCYTO). The cell suspension concentration was reconstituted to 1000 cells per μL in DPBS for DNBelab C4 system library preparation.

*Lung*. The fresh lung samples were cut into 1 cm pieces and washed with 2 ml cold PBS solution 3–5 times. Then the samples were minced into small pieces in PBS solution on a dish with sterilized scissors on ice (within 5 min). About 0.5 g tissue mixture was picked and transferred into three 5 ml centrifuge tubes with 2 ml digestion buffer containing 10 mg/ml Bacillus licheniformis protease (Sigma #P5380), 0.5 mM EDTA in cold DPBS. The tissue digestion was performed at 25 °C with gentle shaking for 30 min. Then the cell suspension from three tubes was mixed for filtering.

*Gill*. The fresh gill samples were washed with 2 ml cold PBS 3–5 times and minced into small pieces on a dish in PBS on ice (within 5 min). Then the gills were transferred to 5 ml centrifuge tubes with 2 ml dissociation enzymes containing 13 Wünsch units/ml liberase TL enzyme mix (Roche #05401020001) in cold DMEM. The remaining steps were the same as the lung tissues described above.

### scRNA-seq library construction and sequencing
Except for two freshwater gill samples, all single-cell RNA-seq libraries were prepared using a DNBelab C Series Single Cell Library Prep Set (MGI, 1000021082)[117]. These included 21 libraries from the lung (eleven libraries of the freshwater group and ten libraries of terrestrialized group) and 27 libraries from the gill (seventeen libraries of the freshwater group and ten libraries of terrestrialized group, libraries) were obtained. Briefly, single-cell suspensions were used for droplet generation, emulsion breakage, bead collection, reverse transcription, and cDNA amplification to generate barcoded libraries. Indexed libraries were constructed according to the manufacturer's protocol. Libraries were constructed and the DNA nanoballs (DNBs) were loaded into the patterned nanoarrays and sequenced on a DNBSEQ-T1 sequencer at the China National GeneBank (Shenzhen, China) with the following sequencing strategy: 41-bp read length for read 1 and 100-bp read length for read 2.

Two freshwater gill libraries (FW-gill9 and FW-gill10) were generated with a 10x Genomics Chromium 3'Reagent Kit instructions (v2) and sequenced on a MGISEQ 2000 sequencer. Gill cells were resuspended in PBS with 0.04% BSA and loaded into a 10x Genomics Chromium cell controller to generate a library for sequencing (28 base pairs for read 1 and 98 for read 2).

### Single-cell RNA-seq data processing
The raw sequencing data were mapped to the West African lungfish genome (https://figshare.com/articles/dataset/The_gff_file_cds_file_and_pep_file_of_the_African_lungfish_genome/13725901)[4], of which gene sets were extended and supplemented using transcriptome data manually, by Cell Ranger v3.0.2 (raw data generated using the Chromium Single Cell 3' v2 kit) and DNBelab C Series scRNA analysis software (MGI)[118] respectively. A gene expression matrix was processed by SoupX[119] to remove cell-free mRNA contamination. Next, we filtered cells with detected gene numbers of less than 200 or more than 5,000

and predicted doublets using DoubletFinder[120], removing 3% of the most similar cells.

After basic qualification, we mainly used Seurat v4[35] for further analysis. For each library, we performed normalization, scaling, and identified variable genes using the function *SCTransform* and then integrated all data from the same tissues. We set the normalization.method as "LogNormalize" for lung data and "SCT" for the gill data when running the *FindIntegrationAnchors* function. Next, we ran principal component analysis (PCA) on integrated objects using *RunPCA* with "npcs = 30" and reduced dimensions with the Uniform Manifold Approximation and Projection (UMAP) method[36] by calling *RunUMAP* (reduction = "pca", dims = 1:30). Regarding clustering, we set the resolution as 0.5 in *FindClusters* and identified DEGs in each clusters by running *FindAllMarkers* with assigned criteria (min.pct = 0.25, logfc.threshold = 0.25, test.use = "wilcox").

We downloaded three scRNA-seq datasets from previous studies and reanalyzed them using the same pipeline described above. Of note, data from multiple libraries were integrated with the "SCT" method. The resolution to identified clusters was Atlantic salmon gill (0.5), mouse lung (0.6), and human lung (0.8). Next, we used the provided cell markers in each paper to annotate cell types and got three cell atlases: human lung (17,208 cells)[90], mouse lung (7,549 cells)[90], and Atlantic salmon gill (18,991 cells)[92] (Supplementary Fig. 5a–c). Additionally, we used the original cell type definition results for the zebrafish gill (11,667 cells) and zebrafish swim bladder (15,237 cells) downloaded from the ZCL database[91] for comparison with the other species.

## Collection of ortholog genes across species

We used diamond v0.9.14.115[121] to perform BLAST analysis between lungfish gene sequence and protein sequence of other species. The genome assemblies used were human GRCh38.p12, mouse GRCm38.p6, zebrafish GRCz11, and Atlantic salmon ICSASG_v2. We set the BLAST parameters as: -f 6 --sensitive --evalue 1e-5 -p 1. We retained matches with sequence identity greater than 50% and denoted genes with the highest identity one-to-one orthologous.

We also mapped the West African lungfish genome to the eggnog database using emapper-2.1.3[122]. We set employed the parameters -m diamond --cpu 2 --database euk --target_orthologs one2one. All the ortholog genes can be found in Supplementary Data 12.

## Gene enrichment analysis

We used the *GSEA* function in clusterProfiler v3.14.3[123] to perform enrichment analysis. The input genes were calculated using Seurat *FindAllMarkers*, retaining genes with *P*-values less than 0.05 and converting into human gene symbols using the ortholog data set generated above. Reference gene sets (cell type signature gene sets,C8) were downloaded from Molecular Signatures Database (MSigDB)[124]. The web tool Enrichr[125–127] was used to further evaluate genes.

## Cell-cell communications

We extracted the metadata information and normalized the gene expression matrix from the integrated datasets of lung and gill. Then the gene names were converted into human gene symbols with one-to-one orthologous. Next, we used CellPhoneDB[65] to infer the cell-cell communications with "cellphonedb method statistical_analysis cellphonedb_meta.txt cellphonedb_count.txt --countsdata = gene_name --threads 8". We retained ligand-receptor pairs with a *P*-value < 0.05.

## Cross-species analysis

We subset the control group from integrated dataset of lungfish lung and gill. We first used the count matrix and normalized to compare similarities (see FC_mtx.r in the GitHub repository associated with this manuscript) and differences between species with one-to-one orthologous[27]. We selected the cell type links between species with top 5% KLD values (threshold.quantile = 0.95) and plotted violins with feature_in_thrs = 1.3.

## Statistics and reproducibility

We classified differentially expressed genes (DEGs) of the West African lungfish lungs and gills between freshwater and terrestrialization condition as follows: threshold value: *P*-value less than 0.05 and absolute log$_2$-fold change greater than 0.25. In cross-species comparisons, we retained DEGs with a *P*-value less than 0.05 and absolute log$_2$-fold change greater than 1. DEGs were obtained using the *FindMarkers* function of Seurat and the quotient of normalized data of the two experiment condition.

The H&E and FISH slides were repeated independently 3–5 times. The sequencing libraries for each tissue sample were repeated at least two times. The tissue from different experiment animals was repeated at least two times. All experiments were repeated independently and had consistent conclusions. The experiments were randomized, and the investigators were blinded to allocation during experiments and outcome assessment. No statistical method was used to predetermine the sample size. No data were excluded from the analyses.

## In situ hybridization

Fluorescence in situ hybridization (FISH) was conducted by the previous protocol[4]. The lung and gill filament were freshly sampled from estivated African lungfish and immediately fixed with 4% paraformaldehyde (Servicebio, #G1113) above 12 h. The tissues were dehydrated by successive steps in an ice cold 70%, 90%, 90% (v/v) ethanol (Sinopharm, #100092683) series, for 2 min per step, followed by a 5 min dehydration in neat ethanol, then embedding in paraffin. The paraffin was sliced through the Pathologic microtome (Leica, # RM2016) (5 μm), roasted for 2 h at 62 °C. Then the sample slices passed in BioDewax and Clear solution (Servicebio, G1128), pure ethanol (SCRC, #100092683) and DEPC water. The slices were boiled in the repairing solution for 10 min, naturally cooled, digested by proteinase K (Servicebio, # G1234) (20 mg/mL) at 37 °C for 15 min, washed in pure water and washed in PBS (Servicebio, #G0002) for 5 min each with three times. After digestion, the sample slices were added prehybridization solution (Servicebio, China) and incubated for 1 h at 37 °C, then the pre-hybridization solution was removed. For hybridization, the sample was incubated in hybridization solution (Servicebio, China) including prepared probe for marker gene (Supplementary Data 13) overnight at 40 °C. Then the slides were washed sequentially in 2 × SSC (Servicebio, China) for 10 min at 37 °C, 1 × SSC two times for 5 min each at 37 °C, 0.5 × SSC for 10 min at room temperature. The sample slides were incubated with DAPI (Servicebio, #G1012) for 8 min in the dark. Sections were imaged with an Eclipse Ci fluorescence microscope (Nikon, Japan). CaseViewer (version 2.4.0.) was used for the image screenshot and resolution adjustment.

## Reporting summary

Further information on research design is available in the Nature Portfolio Reporting Summary linked to this article.

## Data availability

The single cell RNA sequencing raw data and matrix data generated in this study have been deposited in the CNGB Sequence Archive (CNSA) of China National GeneBank DataBase (CNGBdb) under accession code CNP0003631. The processed scRNA-seq data and scRNA-seq raw data are available in the GEO database under accession code GSE240094. The cell type annotation file and updated gene annotation file are available in the Figshare [https://doi.org/10.6084/m9.figshare.23790057]. The West African lungfish genome and annotation file are available in the Figshare [https://figshare.com/articles/dataset/

The_gff_file_cds_file_and_pep_file_of_the_African_lungfish_genome/ 13725901]. The published scRNA-seq data of human and mouse lung used in this study are available in the GEO database under accession code GSE133747. The zebrafish gill and swim bladder scRNA-seq data used in this study are available in the GEO database under accession code GSE130487. The Atlantic salmon gill scRNA-seq dataset used in this study are available in the GEO database under accession code GSE166686. Genes related to phenotype of zebrafish swim bladder from ZFIN database used in this study are available at [http://zfin.org/ search?category=Anatomy+%2F+GO&q=swim+bladder]. Reference gene sets from MSigDB for GSEA analysis are available at [https://www. gsea-msigdb.org/gsea/msigdb/human/collections.jsp#C8]. All the other related data and file are available in the supplementary files and Source Data file.

## Code availability

The code is available at Github (https://github.com/BGI-Qingdao/ Lungfish_scRNA_data_analysis) and Zenodo (https://doi.org/10.5281/ zenodo.8183003). Source data are provided with this paper.

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

## Acknowledgements

This work was supported by a grant from The General Program (Key Program, Major Research Plan) of National Natural Science Foundation of China (No. 32170439, to G.Y.F.), the major scientific and technological projects of Hainan Province (ZDKJ2019011), the Specially-appointed Professor Program of Jiangsu Province (to I.S.), the Jiangsu Foreign Expert Bureau (to I.S.), and the Jiangsu Provincial Department of Technology (grant JSSCTD202142 to I.S.). This work was also supported by the Guangdong Provincial Key Laboratory of Genome Read and Write (No. 2017B030301011). We thank the China National Gene Bank for the sequencing data archiving and storage.

## Author contributions

G.F., R.Z., Q.L., and I.S. conceived the study. M.Z. purchased the specimens. R.Z, Q.L., Y.Q., and M.Z induced the West African lungfish terrestrialization and took care of the specimens daily. Y.L. and Z.Y. performed sample collection. S.P., Y.Z., Y.Q., N.Z., and J.M. conducted the scRNA-seq, ISH experiments and sequencing. Q.L., R.Z., and Y.S. performed the quality control, clustering, and integration of the scRNA-seq data. R.Z. and Q.L. annotated the cell type and compared scRNA-seq data among different species. K.W., S.H., provided guidance on lungfish experiment. X.J. provided guidance on lungfish lung alveolar structure. Z.Z., S.L., M.N., X.L., X.X., H.Y., and J.W. contributed to reagents, materials and computing resources. R.Z., Y.Z., and Y.Q. wrote the manuscript with the results from all the authors. G.F., I.S., X.D., and Q.L. revised the manuscript. All authors read and approved the final manuscript.

## Competing interests

The authors declare no competing interests.
