## [Peer Review file · Nature Communications]

REVIEWER COMMENTS

Reviewer #1 (Remarks to the Author):

This is an interesting study that uses for the first time single cell RNA-Seq to uncover cell types at respiratory surfaces of African lungfish. This can be a very useful resource for the evolutionary biology community. The manuscript does a very good job at covering the existing literature pertaining lungfish biology. Although interesting, I have major methodological concerns about the paper and the data interpretation.

- Lines 437-461: the English in all these methods section needs complete editing. The methods are written as if this was a laboratory protocol.

- Tissue collection: animals were not bled before sample collection, rather they were killed by a blow to the head. Thus, both the lungs and the gills should have a lot of blood contaminating the tissue perhaps explaining the high numbers of immune cells identified in the gill single cell data set. Additionally, this should be mentioned in the discussion because obviously authors now have the issue of resident versus circulating immune cells mixed in all their datasets.

- Gill and lung single cell suspension: the methods described are not standard methods for cell isolation from mucosal tissues. Without DTT and enzymatic treatment (collagenase), cells from the lamina propria were likely not released.

- Please include flow cytometry plots of the isolated cells before single cell.

- Please include routine histology (H&E stains) of the tissues used for single cell since it will help understand the subsets identified especially the unknown immune cell types. Add this information to main Figure 1.

- Overall, I am very surprised that the authors did not recover MPO+ granulocytes since these are the most common immune cell in lungfish. Previous studies have identified other markers for lungfish granulocytes such as *cxcr2* and *elane*. Please comment on whether or not these markers were found and if they weren't; this may be due to the method of choice for cell isolation

- The assignment of immune cell subsets is very hand wavy. For instance, the Immune cell_F13A1_SLC37A is mentioned in the text to express *LC37A2* as well as *F13A1* but in Fig. 5b this subset expresses high levels of myosin light chain genes. Please comment on this expression pattern.

- Another example is the NKT cell assignment, since NKT cells have never been described in ectotherms and the evolution of NK cell markers is extremely complex and unknown other than in mammals. Authors should elaborate more on what gene markers made them assign these to NKT cells since truly, this is a big enigma in evolutionary immunology

- Finally, many B cells should be present in these datasets since they are the main cell type of all fish mucosal surfaces and immunoglobulin genes have been well investigated in lungfish. The B cell subsets

shown in figure 2g do not show any igh genes. Please also comment whether B cell numbers changed upon terrestrialization.

- I have searched for CNP0003631 in the <https://db.cngb.org> but the dataset cannot be accessed. Please ensure that data is readily available for others to download.

Reviewer #2 (Remarks to the Author):

The manuscript by Zhang et al. reports a single cell transcriptome atlas of the lung and gills of the West African lungfish. Lungfish are the closest relatives of the tetrapods and can provide information on the evolutionary-developmental processes of the water to land transition. One important aspect of this process is the change in respiration from the aqueous environment to air and from gills to lungs. Thus, the here presented atlas of both adult organs from the bimodal lungfish is an important archive for further morphological, physiological, and developmental-biological studies. It is the first time that such a dataset is reported. Hence, this is a study of considerable novelty. Previous work comparing lung and gill cell types were exclusively morphological.

Like in most similar sc transcriptome studies, differences in the expression profiles of cell types allowed also here to identify sub-types that were not recognized before. Most of the sc data are reflecting the earlier anatomical-histological and immunohistochemical data. They also confirm that lungfish have only a single alveolar epithelium and that only the amniotes innovatively evolved two different cell types that make up the epithelium of the alveolae.

The terrestrialization experiment uncovered only a mild effect of this physiologically extremely challenging process on cell type, but more changes on the genome-wide gene expression level. DEGs were all within expectation for the lowered respiration rate.

In summary, this is a dataset that will be of considerable interest to the field. The manuscript is well written, and the conclusions are supported by the data. The relevant literature is appropriately considered.

The generation of the sc transcriptome data and the bioinformatic treatments followed the established standard procedures. The methods are well described. There are now technical problems with this manuscript. For identification of cell types published datasets from human, mouse, salmon, and zebrafish were used. I was wondering whether such an approach – besides identifying the homologous cell types of lungfish – would ignore cell categories that are specific to lungfish.

The enthusiasm for this paper is somehow lowered by the fact that only one terrestrialized and two aestivating individuals were used. This affects in particular the DEGs that were used to infer the cellular and molecular features of lungfish terrestrialization. It would be appropriate to include a note of caution here.

Minor points:

- L 169 – 311 This is a very long section, and it is difficult to read because inferences that deal specifically with terrestrialization and discussion of more general features of lungfish gill and lung (e.g. signalling interactions) are mixed. This should be separated into two sections.
- L 445-461: There seems to be some copy/paste from lab protocol mixed with regular manuscript writing. Adjust to general manuscript style.
- L 504 -512: The URLs from which the datasets were downloaded should be given
- L 565: Are these the same individuals as those used for the sc transcriptome sequencing?
- L 588 – 589: Give the URL how to connect to the Sequence Archive and to access the primary data.

Reviewer #3 (Remarks to the Author):

The West African lungfish is one of 6 still living lungfish species and has the remarkable capacity to change its lifestyle between a dormant “terrestrialized” state and a natural aquatic state. During dormancy the lungfish relies solely on lung breathing whereas it uses a mix of lung and gill breathing in its natural aquatic environment. The interesting phylogenetic position of lungfishes as a sister group to tetrapods makes them an interesting model to study changes that occurred at the transition from water to land. In this study, the authors use scRNA-seq to compare lungs and gills during dormancy with their freshwater counterparts and to look for similarities and differences to the lungs of mammals and the gills and swim bladder of fish. The study reveals many gene expression differences across the respiratory tissues studied which will be a great resource for understanding the changes and challenges for the lungfish during dormancy but also for studying the evolution of the cell types that were essential for the process of the sea-land transition. The primary claim of the study is that there are broad transcriptional similarities between the cell types of the fish swim bladder, the lungfish lung and the mammalian lungs which is not completely surprising as it has been shown that these are most likely homologous organs. The study, however, adds insights to the evolution of these organs at single-cell resolution, sheds light on gene expression differences between the dormant and freshwater state of the organs investigated in the lungfish and reveals potential ligand-receptor based cell type interactions occurring during this process. These findings are noteworthy and supported while other claims – mostly because of the qualitative and not quantitative nature of scRNA-seq, are rather speculative and not sound given the caveats below:

1. Cell type composition comparisons

I assume that lungfish samples are precious and not easily available which might explain why the number of animals used is small. The authors therefore ran all the samples with multiple replicates in parallel which allows to obtain more cells and also to make the scRNA-seq data across different lungfish samples more reliable for DE gene analyses.

However, the replicates do not make cell type ratio estimations more reliable as the authors claim. The way I understand the single-cell preparation process is that the gill and lung samples, respectively, are rather technical replicates with the same cell suspension being loaded on multiple 10x lanes in parallel. It is therefore not surprising that the replicates of the same cell suspension originating from the same lung of a single fresh water animal are very similar in cell type ratio numbers. This only shows that the 10x controller does not introduce any random batch effect during the cell encapsulation. In fact, there is a clear difference between AE-lung5-8 and AE-lung1-4 (Fig. S1C) which I assume belong to animal 1 and animal 2 and thus two separately obtained cell suspensions. However, this information as well as a more detailed explanation of the sampling regime is missing and needs to be added. I do therefore not fully agree with using mean values across all the samples which has to be more critically discussed. In particular, there are almost no goblet and Un_epithelial cells detected in AE lung samples 5 to 8 (similarly AEgill5-8 samples have more T cells). Interpretations based on comparisons of cell type ratios between the AE and FW lung as well as AE and FW gills, respectively, (e.g. Fig. S3 c,d) should therefore be revisited critically if not in parts removed completely.

One obvious preparation bias is the circulating blood during the tissue dissociation in my opinion. How much blood ended up in the dissociation is going to have a huge impact on red blood and immune cell numbers and hence the overall ratio of all other cell types. However, there seems to be not even a washing step of the gills. More details need to be provided in order to evaluate if the authors accounted for such effects. The authors even give an example by themselves in line 219 that calling cell type ratios solely from single-cell data is misleading: "In contrast, a reduction in endothelial cell number is, at first glance, counterintuitive as the lung blood vessels of the West African lungfish are highly vascularized and dilated at the maintenance phase." which is the opposite of the cell type ratio estimated based on scRNA-seq results. It might be an option to evaluate the differences by leaving out cells that are part of the blood. However, I don't know how reliable this distinction is possible for lungfish.

Taken together, if cell type ratio comparisons are made, they have to be at least partially validated by in situ comparisons (e.g. cell counting, literature) and should not solely rely on scRNA-seq cell type ratios. Another option would be to add more independently generated lungfish gill and lung scRNA-seq data points assuming that the biases are at least similar across all the experiments. How much this finally reflects the actual tissue cell type ratios would be, however, still questionable. A more direct validation would be spatial transcriptomic techniques (e.g. 10x Visium). If none of these validations are possible all claims based on cell type abundances should be critically revisited or removed.

2. Staining

The validation of transcriptomic findings by in situ staining methods is commonly used for single-cell transcriptomics and I appreciate that the authors try to validate their findings given the fact that lungfishes are not a routine model organism.

However, a greater magnification is needed to interpret the staining as it would be interesting to see whether the cell type specific genes identified are indeed expressed in separate cells in the tissue as suggested by the scRNA-seq results. With the resolution presented I don't see a clear difference between the staining for *npc2* and *arpc1b* even though the single-cell data suggests that the genes should be found in different cell types. I am not an expert in lung tissue architecture and hence I don't know if these different populations are indeed in close proximity in the lung. Maybe a schematic could help to show that the cell types are expected to be found in the same part of the lungs. Generally, more explanations on how to read the images would be helpful.

3. Cell type evolution

I think that the evolutionary perspective across different taxa ranging from fish to human is one of the most interesting aspects of the study presented. However, the methods used and the data presentation are not fully sound in my opinion.

The interpretation of the cell type similarities is heavily dependent on the right assignment of cell types across the different data sets in the first place. The clusters were obtained unbiasedly and then afterwards assigned based on published marker genes for these data sets (human, mouse and salmon Fig. S5a-c) before similarity calls were made between different clusters. But there are explanations missing why the authors changed the strategy for the zebrafish data set or vice versa. What was the rationale for this and does it have an impact on the cluster similarity assignment?

As a proof for a correct cell type assignment and cluster alignment across the data sets the authors focus only on orthologs in the alveolar cells (Fig. 4b). But what is the pattern for the other cell types that are assigned across the different species? Are the other similarity assignments also meaningful on the gene level i.e. are there sets of transcription factors that allow a clear assignment to a certain cell lineage. In other words, I would like the authors to focus on the similarities rather than the differences which are mostly the focus of Figure 4. This would validate in parts the cluster assignments and is for the interpretation of evolutionary directions essential.

Legend and method information are not available in a straightforward way. It is not clear to me what the grey lines in Fig. 4a are telling and based on which criteria they are drawn. Generally, a brief introduction of the method used to compare across species would be helpful and is currently missing in the text. The method was originally established for comparing 3 cnidarian whole body cell type atlases. A more detailed validation of the inferred cell type similarities by shared gene expression programs would not only improve the value of the comparisons but would also prove that the method applied not only works for a broad lineage assignment of cell types at the whole-body scale for the rather simple body plan of cnidarians (cnidocytes, neurons and gastrodermis) but also for the assignment of cell types on a more fine-grained level between lungfish and human lungs. I believe that critically evaluating the performance of the similarity assignment tool would be also interesting to other researchers working on scRNA-seq cross species comparisons.

In addition, the total number of orthologs identified is missing and should be added. If the number is large enough, can the datasets be integrated into one matrix to make the cell type assignment more coherent? Another idea in line with this could be to transfer the label from one data set to the other which is a function in the Seurat package which the authors are anyway using. In case the number of one-to-one orthologs is too small could the authors maybe use tools that are not based on this strict criterion such as SAMap to identify similar cell types? These are just some ideas that could also help to make the similarity calls between cell types more robust and reliable.

4. Unsupported claims about immunity

While mostly descriptive work, the authors are often pushing for speculative interpretations of the scRNA-seq results which are not supported in the current form of the manuscript. Examples are for instance found on lines 147-152, 197, 326, 355 or 372 and should be removed or moved in parts to the discussion section.

To provide an example: “However, the human lung has developed additional functions, including an ostensibly stronger immune system”, is from an evolutionary perspective not appropriate and also not supported by simply finding more different cell types in the human lung scRNA-seq data.

Minor points

- Figure number assignments in the text should be double-checked, e.g.

S4 to S5 line 510, Line 276/77 Fig. S3 assignment seems not correct

- The color legend in Fig. 4c should be adjusted to correspond to the color panel in Fig. S5a for easier reading.
- Increasing small font sizes in figures e.g. scale bar for in situ plots.

- Reference 5 and 6 are duplicated
- The title is a bit misleading as the single-cell study focuses only on lungs and gills and not the whole lungfish as “A single-cell transcriptome atlas of the West African lungfish ...” might suggest.
- More details in the figure captions are needed to understand the figures at the first glance.
- I would highly appreciate when the R codes used would be made publicly available (e.g. gihub). The methods section is in its current state often not helpful in understanding which exact commands were used when the authors refer to custom scripts.
- I think that the in parts extended explanations about other lungfish species in the introduction are distracting from the actual species used in the study. I see potential to streamline the introduction when focusing more on the characteristics of the West African lungfish only.
- In contrast, the discussion has rather a summary character and could potentially be extended by moving in parts interpretations from the results section to this section.

Point-to-point responses

Reviewer #1 (Remarks to the Author):

Comment 1.1: This is an interesting study that uses for the first time single cell RNA-Seq to uncover cell types at respiratory surfaces of African lungfish. This can be a very useful resource for the evolutionary biology community. The manuscript does a very good job at covering the existing literature pertaining lungfish biology. Although interesting, I have major methodological concerns about the paper and the data interpretation.

Response

We thank the reviewer for summarizing our study and highlighting its significance, and providing us with many constructive suggestions that allowed us to improve our manuscript significantly. Our point-by-point responses are detailed below.

Comment 1.2: Lines 437-461: the English in all these methods section needs complete editing. The methods are written as if this was a laboratory protocol.

Response

Thank you for your careful review. The methods section has been thoroughly revised and edited by a native English speaker, so we hope it can meet the journal's standards. (Lines 446-504).

Comment 1.3: Tissue collection: animals were not bled before sample collection, rather they were killed by a blow to the head. Thus, both the lungs and the gills should have a lot of blood contaminating the tissue perhaps explaining the high numbers of immune cells identified in the gill single cell data set. Additionally, this should be mentioned in the discussion because obviously authors now have the issue or resident versus circulating immune cells mixed in all their datasets.

Response

This is a very helpful suggestion. To evaluate the impact of the blood contamination, we performed additional experiments for two gills and one lung after removing as much blood as possible by performing additional washing (3-5 times with cold PBS

solution) (**Figure R1.1**). Also, we tried to perfuse with PBS but failed due to the thinness of the lungfish's lung ^{1 2}.

Figure R1.1. Removal of blood contamination. Representative photograph of (A) lung and (B) gill sample before and after additional washing steps to remove blood.

For the number of immune cell comparison between the original and new protocol, we found the ratio of immune cells in lung samples to be similar (2.50% to 4.95%), while the ratio of immune cells in gill samples was notably reduced from 56.96% to 13.28% (**Figure R1.2**). We are very grateful for the reviewer's suggestion.

Figure R1.2. Effect on removal of blood on cell type assignment. Uniform Manifold Approximation and Projection (UMAP) of before (left) and after (right) additional washing steps to remove blood in the gill (A) and lung (B). Cells are color-coded by cluster cell type.

We integrated the results of the two experiments and obtained 45,427 lung cells and 70,780 gill cells, which were assigned into 14 and 21 cell types, respectively. Compared to our initial manuscript submission, the cell type annotation became clearer and more precise because there was a significant increase in the number of cells (29,640 to 45,427 in lung, 19,385 to 70,780 in gill) (**Figure R1.3**).

Figure R1-3 Sequencing of additional lungfish samples detects more cells and largely preserves the cell type assignments. Uniform Manifold Approximation and Projection of (A) gill and (B) lung cell types our initial submission (left panel) and a dataset including additional scRNA-seq samples. Cells are color-coded by cluster cell type.

Our new results illustrate the reliability of biological and technical replications of the assigned cell types. According to the reviewer's suggestion, we mentioned this issue in the discussion of the revised manuscript. **(Lines 384-394)**

Comment 1.4: *Gill and lung single cell suspension: the methods described are not standard methods for cell isolation from mucosal tissues. Without DTT and enzymatic treatment (collagenase), cells from the lamina propria were likely not released.*

Response

We agree with the reviewer about the importance of enzymatic treatment. To obtain a better cell suspension, we did a lot of preliminary experiments based on published tissue disaggregation and enzymatic dissociation methods for gill and lung^{3 4 5}. After PBS washes to remove blood and mucus, we treated gill and lung tissue using two frequently-used enzymes, Liberase TL and *Bacillus licheniformis* protease, followed by an assessment of cell viability (see *Methods*).

Lungfish gill tissue was dissociated using Liberase TL (Roche #05401020001), an enzyme cocktail containing collagenase I and II. We found that Liberase TL did not dissociate lung tissue well and instead used a cold-active protease from *Bacillus licheniformis* (Sigma, #P5380), an enzyme widely used in single-cell sequencing

studies^{6,7} – including a large-scale study of the human airway⁶. We appreciate that the *B. licheniformis* protease method likely resulted in poor isolation of lamina propria cell types (i.e., primarily lymphocytes) and have added the following to the main text: “Please note that Liberase digestion of the lung resulted in poor cell viability and we instead used the general purpose *Bacillus licheniformis* protease (see⁶), meaning that few cells from the lamina propria were likely dissociated and that immune cells (e.g., lymphocytes) in our lung dataset are likely solely intraepithelial in origin. Future studies should further optimize the isolation of the lungfish lung compartments to investigate the immune cell repertoire of the lungfish lung (e.g., see^{7,8})”.

Comment 1.5: Please include flow cytometry plots of the isolated cells before single cell.

Response

According to the reviewer’s suggestion, we provide the flow cytometry plots of the isolated cells to illustrate the tissue disaggregation and enzymatic dissociation results (**Figure R1.4**). In our opinion, good cell dissociation is observed.

A

B

Figure R1.4. Flow cytometry plots of the isolated cells of lungfish tissues. (A) Flow cytometry plots of isolated cells of lungfish lungs. (B) Flow cytometry plots of isolated cells of lungfish gills.

Comment 1.6: *Please include routine histology (H&E stains) of the tissues used for single cell since it will help understand the subsets identified especially the unknown immune cell types. Add this information to main Figure 1.*

Response

According to the reviewer's suggestion, we performed H&E staining of lungfish lungs and gills. Because the limited cellular morphology description of African lungfish cells, we only found data on African lungfish blood cell, skin cell components, and mucosal lymphoid organs ^{9 10 11 12} and labeled these cell types on the H&E stains plots. We show alveolar epithelial cells and macrophages near the lungfish lung alveolar structure, as well as lymphoid nodes and macrophages on the lungfish gill filament (**Figure R1.5**). We agree with the reviewer about using routine histology to help distinguish unknown cell types. However, due to the small number of unknown immune cell clusters and the lack of morphological information of these unknown immune cells, we are unable to precisely point them out in the H&E images.

Figure R1.5. The H&E stain of African lungfish lungs and gill filament. (A) The overall H&E staining image of lungfish lungs. (B) The Detail View Profile of the red box (I) in (A). The ● represents alveolar epithelial cells; ◆ macrophage cells. (C) The Detail View Profile of the red box (II) in (A). Red circle represents the lymphoid nodes. (D) The overall H&E staining image of lungfish gills. (E) The Detail View Profile of the red box(I) in (D). The ◆ denotes macrophage cells, a red circle lymphoid node.

Comment 1.7: Overall, I am very surprised that the authors did not recover MPO+ granulocytes since these are the most common immune cell in lungfish. Previous studies has identified other markers for lungfish granulocytes such as *cxcr2* and *elane*. Please comment on whether or not these markers were found and if they weren't; this may be due to the method of choice for cell isolation

Response

According to the reviewer's suggestion, we manually identified eight orthologous genes of *MPO*, *cxcr2*, and *elane* in the lungfish genome (**Table R1.1**). We then examined the expression pattern of these eight genes in gill and lung samples.

We found that the neutrophil elastase (*elane*) is highly expressed by the gill "GZMH_high" cell type. Combining the other markers, including *GZMH*, *ABCB1A*, *MEF2C* and *ALOX5AP* (**Extended Data Fig. 1a-b**), we classified the "GZMH_high" cell type as "neutrophil" in the revised manuscript. Lung macrophages show a high expression level of chemokine receptor gene *cxcr2*, consistent with the previous study (**Figure R1.6**)¹³. As for MPO+ granulocytes, we note it was described in the skin epidermis and dermis of lungfish^{10 11} but is not expressed in our gill and lung data, suggesting that the reported MPO+ granulocytes reflect tissue-specific expression.

Table R1.1. Orthologous genes of granulocytes cell markers.

Symbol	Lungfish genes
MPO	Gene2749, Gene5870, LZX_CL10601.Contig5-All-g
CXCR2	Gene15617, Gene15618, Gene15619
ELANE	Gene5453, Gene5616

Figure R1.6. The expression patterns in of lungfish granulocytes markers. (A) Bubble plot of typical granulocytes markers shows the expression pattern across cell types of lungfish gills. Circle size reflects the percentage of cells within a cell type which express the specific genes. The shades of color show average expression levels of specific gene. (B) Bubble plot of typical granulocytes markers shows the expression pattern across cell types of lungfish lungs. Annotated as in **Fig.R1.6**.

Comment 1.8: *The assignment of immune cell subsets is very hand wavy. For instance, the Immune cell_F13A1_SLC37A is mentioned in the text to express LC37A2 as well as F13A1 but in Fig. 5b this subset expresses high levels of myosin light chain genes. Please comment on this expression pattern.*

Response

We agree with the reviewer about the complexity of immune cell classification. We carefully inspected the highly expressed genes of the “immune cell_F13A1_SLC37A” cell cluster. We found that this cluster was associated with platelets because most of the top expressed genes were associated with associated functions (i.e., the myosin light chain) – which became clearer using the integrated data set (**Figure R1.7**). For example, the top two genes highly expressed in this cluster, *F13A1* (encodes the coagulation factor XIII A subunit) and *TIMP4* (tissue inhibitor of metalloproteinases 4), are inhibitors of matrix metalloproteinases (MMP) in human platelets. Another top-ranked gene, *RGS1* (regulator of G-protein signaling 1), regulates platelet activation and myosin-dependent contraction.

The text clarifies that “immune cell_ F13A1_SLC37A” is likely a platelet cell type. After we added new data (additional samples), the number of cells of this cell type increased from 123 to 189, increasing the cell classification accuracy.

Figure R1.7. The cell features of Immune cell_ F13A1_SLC37A in African lungfish gills data.

(A)Violin plot of representative markers of Immune cell_ F13A1_SLC37 cell cluster.

Comment 1.9: Another example is the NKT cell assignment, since NKT cells have never been described in ectotherms and the evolution of NK cell markers is extremely complex and unknown other than in mammals. Authors should elaborate more on what gene markers made them assign these to NKT cells since truly, this is a big enigma in evolutionary immunology.

Response

We agree with the reviewer that the assignment and the evolution of NK and NKT cells are complex. The granule exocytosis and FasL/Fas interaction are the two mechanisms that cytotoxic T lymphocytes and NK cells lyse target cells in mammals¹⁴. Our results reveal that the NK cell markers Granzyme M (*GZMM*) and Granzyme A (*GZMA*) and the Fas ligand gene *FASLG* are highly expressed by the NK like cells

of lungfish gills. The GSEA enrichment result of this cell cluster showed that the top annotated cell types are lymphoid cells and NK cells (**Figure R1.8B**). The NKT cell assignment included marker gene expression of NK cell and T cell receptor (TCR), but TCR (*CD3D*) is lowly expressed in this cell type in lungfish gills (**Figure R1.8A**). Thus, we proposed that there are NK-like cells and not NKT cells in the lungfish gill. Based on our new integrated dataset (i.e., additional biological replicates), we believe we have identified T cells, CD8+T cells, and NK-like cells in the lungfish gill (**Figure R1.8A**).

Figure R1.8. The cell features of NK cell in African lungfish gills data. (A) Bubble plot of typical markers of NK cell, T cell and CD8+T cell shows the expression pattern across cell types of lungfish gills. Annotated as in Fig.R1.4A. (B) Bubble plot of top GSEA terms of DEGs of NK cell-like in the gill. GSEA denotes Gene Set Enrichment Analysis; DEG, differentially expressed gene. The color gradient in the legend represents the normalized enrichment score, circle size and the color reflect the $-\log_{10}(\text{pvalue})$.

Comment 1.10: Finally, many B cells should be present in these datasets since they are main cell type of all fish mucosal surfaces and immunoglobulin genes have been well investigated in lungfish. The B cell subsets shown in figure 2g do not show any igh genes. Please also comment whether B cell numbers changed upon terrestrialization.

Response

According to the reviewer’s suggestion, we carefully checked the B cell cluster in our dataset. We used marker gene expression and GSEA enrichment to identify B cells. We found that the B cell receptor (*CD22* and *CD81*) and the transcription factor *SPI1* (roles in B-lymphoid cell development¹⁵) show high expression in this cluster (**Figure R1.9A**). Moreover, the top GSEA terms are enriched for B cell functions (**Figure R1.9B**). Thus, we classified 1,096 cells into a B cell cluster, only 1.55% of the lungfish gill cell atlas (70,780 cells). Comparing the B cell number between freshwater and terrestrialized lungfish gills, there are 270 (2.16%) B cells in terrestrialized gills and 826 (1.42%) in freshwater gills. However, the results of cell number and cell proportion comparisons were inconsistent. Combined with suggestions from reviewer#2, we found that there were individual differences in the comparison of cell proportion. Thus, we removed all the results related to comparing cell number and proportion in our revised manuscript.

To check the expression level of *Igh* genes, we obtained 152 immunoglobulin genes with high sequence identity, including 117 *Igh* genes in the lungfish genome. We found that three *Igh* genes and *IGBP1* showed significantly reduced expression during terrestrialization (**Figure R1.9C**). We added the results of these four *Ig* genes into the revised Figure 2g.

Figure R1.9. The cell features of B cell_like in African lungfish gills data. (A) Bubble plot of typical markers of B cell_like shows the expression pattern across cell types of lungfish gills. Annotated as in Fig.R1.4A. (B) Bubble plot of top GSEA terms of DEGs of B cell_like in the gill. Annotated as in Fig.R1.6B. (C) Bubble plot of Igs genes shows the expression levels in B cell_like of gills between freshwater lungfish and aestivated lungfish.

Comment 1.11: *I have searched for CNP0003631 in the <https://db.cngb.org> but the dataset cannot be accessed. Please ensure that data is readily available for others to download.*

Response

Apologies, the data is publicly available (<https://db.cngb.org/search/?q=CNP0003631+>).

Reviewer #2 (Remarks to the Author):

Comment 2.1: *The manuscript by Zhang et al. reports a single cell transcriptome atlas of the lung and gills of the West African lungfish. Lungfish are the closest relatives of the tetrapods and can provide information on the evolutionary-developmental processes of the water to land transition. One important aspect of this process is the change in respiration from the aqueous environment to air and from gills to lungs. Thus, the here presented atlas of both adult organs from the bimodal lungfish is an important archive for further morphological, physiological, and developmental-biological studies. It is the first time that such a dataset is reported. Hence, this is a study of considerable novelty. Previous work comparing lung and gill cell types were exclusively morphological.*

Like in most similar sc transcriptome studies, differences in the expression profiles of cell types allowed also here to identify sub-types that were not recognized before. Most of the sc data are reflecting the earlier anatomical-histological and immunohistochemical data. They also confirm that lungfish have only a single

alveolar epithelium and that only the amniotes innovatively evolved two different cell types that make up the epithelium of the alveolae.

The terrestrialization experiment uncovered only a mild effect of this physiologically extremely challenging process on cell type, but more changes on the genome-wide gene expression level. DEGs were all within expectation for the lowered respiration rate.

In summary, this is a dataset that will be of considerable interest to the field. The manuscript is well written, and the conclusions are supported by the data. The relevant literature is appropriately considered.

Response

We would like to thank the reviewer for the positive comments, and also all the constructive suggestions. We have revised the manuscript according to your suggestions and we think the revised manuscript was thus greatly improved.

Comment 2.2: *The generation of the sc transcriptome data and the bioinformatic treatments followed the established standard procedures. The methods are well described. There are now technical problems with this manuscript. For identification of cell types published datasets from human, mouse, salmon, and zebrafish were used. I was wondering whether such an approach – besides identifying the homologous cell types of lungfish – would ignore cell categories that are specific to lungfish.*

Response

We agree with the reviewer that identifying specific cell categories in non-model species is a big challenge. We carefully identified the cell types by multiple methods in our dataset. Firstly, we identify the lungfish cell types using published markers from model animals, a widely used bioinformatics method. Secondly, we performed Gene Set Enrichment Analysis (GSEA) to confirm the function of each cell type. Thirdly, we further confirmed selected cell types by FISH and H&E experiments. Finally, we classified 20/21 and all well-annotated cell types of gills and lung, respectively. As mentioned by the reviewer, one cell type in the gill (CTSG_high) could not be classified and is likely specific to lungfish. In the subsequent analysis,

we did not focus on this cell type because it only contains a very low cell number (193) but it certainly warrants study using more sensitive future experimental methods.

Comment 2.3: *The enthusiasm for this paper is somehow lowered by the fact that only one terrestrialized and two aestivating individuals were used. This affects in particular the DEGs that were used to infer the cellular and molecular features of lungfish terrestrialization. It would be appropriate to include a note of caution here.*

Response

We agree and have added three lung and seven gill libraries from two freshwater individuals in our revised analysis. We integrated the results of two batches and obtained 70,780 and 45,427 cells, assigned into 21 and 14 cell types in the gill and lung, respectively. Compared to our submission dataset, although there was a significant increase in the numbers of cells (19,385 to 70,780 in gill; 29,640 to 45,427 in lung), there was no apparent change in cell types identified (**Figures R1.2 and R1.3**) (although we were better able to assign marker genes and improve differential gene expression analysis – see earlier responses and below). These results illustrate the reliability of biological and technical replications of cell types.

The DEGs identified in our new dataset overlapped well with our earlier dataset. For example, 92.22% of DEGs in gill ionocytes during terrestrialization overlapped. As expected, increasing our sample number also allowed us to more reliably identify additional DEGs, including genes related to lung immune function such as *HCLS1* (lymphocyte trafficking protein), *SPN* (leukosialin; also known as CD43), and *TNFAIP8L2* (TNF Alpha Induced Protein 8 Like 2) (Fig.2c).

Minor points:

Comment 2.4: - L 169 – 311 *This is a very long section, and it is difficult to read because inferences that deal specifically with terrestrialization and discussion of*

more general features of lungfish gill and lung (e.g. signalling interactions) are mixed. This should be separated into two sections.

Response

We thank the reviewer's suggestion. We have separated this part into the results of gill and lung sections. The detailed revision is in the in the manuscript. (**Lines 171-292**)

Comment 2.5: - L 445-461: *There seems to be some copy/paste from lab protocol mixed with regular manuscript writing. Adjust to general manuscript style.*

Response

Thank you for your careful review. The methods section has been thoroughly revised by a native English speaker (**Lines 446-504**).

Comment 2.6: - L 504 -512: *The URLs from which the datasets were downloaded should be given*

Response

We thank the reviewer's suggestion on the data availability. We listed the URLs of downloaded data (**Table R2.1**) in **Supplementary Data 7**.

Table R2.1. URLs of downloaded single cell scRNA-seq data.

Species-Tissue	URLs
Human and mouse lung	https://www.ncbi.nlm.nih.gov/geo/query/acc.cgi?acc=GSE133747
Zebrafish gill and swim bladder	https://www.ncbi.nlm.nih.gov/geo/query/acc.cgi?acc=GSE130487
Atlantic salmon gill	https://www.ncbi.nlm.nih.gov/geo/query/acc.cgi

Comment 2.7: - L 565: *Are these the same individuals as those used for the sc transcriptome sequencing?*

Response

We used different samples for *in situ* hybridization and single-cell transcriptome sequencing. For the *in situ* hybridization, we collected tissues and put them into stationary liquid to keep the morphology as intact as possible before paraffin embedding. For single-cell transcriptome sequencing, we put whole fresh tissues into cold PBS solution and performed tissue disaggregation and enzymatic dissociation. Therefore, using the same individuals to perform these two experiments is difficult.

Comment 2.8: - L 588 – 589: Give the URL how to connect to the Sequence Archive ant to access the primary data.

Response

Thanks. We have revised the releasing time of data publicly available (<https://db.cngb.org/search/?q=CNP0003631+>).

Reviewer #3 (Remarks to the Author):

Comment 3.1: *The West African lungfish is one of 6 still living lungfish species and has the remarkable capacity to change its lifestyle between a dormant “terrestrialized” state and a natural aquatic state. During dormancy the lungfish relies solely on lung breathing whereas it uses a mix of lung and gill breathing in its natural aquatic environment. The interesting phylogenetic position of lungfishes as a sister group to tetrapods makes them an interesting model to study changes that occurred at the transition from water to land. In this study, the authors use scRNA-seq to compare lungs and gills during dormancy with their freshwater counterparts and to look for similarities and differences to the lungs of mammals and the gills and swim bladder of fish. The study reveals many gene expression differences across the respiratory tissues studied which will be a great resource for understanding the changes and challenges for the lungfish during dormancy but also for studying the evolution of the cell types that were essential for the process of the sea-land transition. The primary claim of the study is that there are broad transcriptional similarities between the cell types of the fish swim bladder, the lungfish lung and the*

mammalian lungs which is not completely surprising as it has been shown that these are most likely homologous organs. The study, however, adds insights to the evolution of these organs at single-cell resolution, sheds light on gene expression differences between the dormant and freshwater state of the organs investigated in the lungfish and reveals potential ligand-receptor based cell type interactions occurring during this process. These findings are noteworthy and supported while other claims – mostly because of the qualitative and not quantitative nature of scRNA-seq, are rather speculative and not sound given the caveats below:

Response

We thank the reviewer for reading our paper carefully and providing constructive comments. We have revised the manuscript carefully according to your constructive suggestions, and we think the revised manuscript was thus greatly improved.

Meanwhile, we also carefully checked the whole manuscript to avoid other missing information during the revision. Our point-by-point responses are detailed below.

1. Cell type composition comparisons

Comment 3.2: *I assume that lungfish samples are precious and not easily available which might explain why the number of animals used is small. The authors therefore ran all the samples with multiple replicates in parallel which allows to obtain more cells and also to make the scRNA-seq data across different lungfish samples more reliable for DE gene analyses.*

Response

Sampling, as well as terrestrialization experiments, is indeed tricky. However, we were able to obtain additional samples during the revision. Please see responses 1-3 and 2-3 above.

Comment 3.3: *However, the replicates do not make cell type ratio estimations more reliable as the authors claim. The way I understand the single-cell preparation process is that the gill and lung samples, respectively, are rather technical replicates*

with the same cell suspension being loaded on multiple 10x lanes in parallel. It is therefore not surprising that the replicates of the same cell suspension originating from the same lung of a single fresh water animal are very similar in cell type ratio numbers. This only shows that the 10x controller does not introduce any random batch effect during the cell encapsulation. In fact, there is a clear difference between AE-lung5-8 and AE-lung1-4 (Fig. S1C) which I assume belong to animal 1 and animal 2 and thus two separately obtained cell suspensions. However, this information as well as a more detailed explanation of the sampling regime is missing and needs to be added. I do therefore not fully agree with using mean values across all the samples which has to be more critically discussed. In particular, there are almost no goblet and Un_epithelial cells detected in AE lung samples 5 to 8 (similarly AEGill5-8 samples have more T cells). Interpretations based on comparisons of cell type ratios between the AE and FW lung as well as AE and FW gills, respectively, (e.g. Fig. S3 c,d) should therefore be revisited critically if not in parts removed completely.

Response

We appreciate these constructive comments. As mentioned by the reviewer, we used the same cell suspension for loading the lanes of each individual. Whole fresh lung and gill tissues were washed in cold PBS solution. Then, we obtained cell suspensions by picking 0.5g homogeneous tissue mixture for disaggregation and dissociation because of the big size of the lung tissue. Each cell suspension was separated into three parts to capture more cells and reduce technical batch effects. A more detailed method description has been added to the manuscript (**Lines 446-504**).

The cell type ratio comparative analysis found that the cell type ratio between the two biological replication experiments of AE5-8 and AE1-4 are different. Since the same staff performed each experimental step, the differences between the replication experiments are likely due to differences between the sampled lungfish. Due to the challenge of unifying the genetic and physiological background of the non-model

lungfish samples, we decided to exclude the comparative analysis of cell ratios from the revised manuscript.

Comment 3.4: *One obvious preparation bias is the circulating blood during the tissue dissociation in my opinion. How much blood ended up in the dissociation is going to have a huge impact on red blood and immune cell numbers and hence the overall ratio of all other cell types. However, there seems to be not even a washing step of the gills. More details need to be provided in order to evaluate if the authors accounted for such effects. The authors even give an example by themselves in line 219 that calling cell type ratios solely from single-cell data is misleading: “In contrast, a reduction in endothelial cell number is, at first glance, counterintuitive as the lung blood vessels of the West African lungfish are highly vascularized and dilated at the maintenance phase.” which is the opposite of the cell type ratio estimated based on scRNA-seq results. It might be an option to evaluate the differences by leaving out cells that are part of the blood. However, I don’t know how reliable this distinction is possible for lungfish.*

Response

Thank you for raising this important point. We introduced additional washing steps to remove blood (please see response 1-3). These results suggest that washing alters the proportion of cells. Thus, we removed the misleading description about the cell ratio from our revised manuscript.

Comment 3.5: *Taken together, if cell type ratio comparisons are made, they have to be at least partially validated by in situ comparisons (e.g., cell counting, literature) and should not solely rely on scRNA-seq cell type ratios. Another option would be to add more independently generated lungfish gill and lung scRNA-seq data points assuming that the biases are at least similar across all the experiments. How much this finally reflects the actual tissue cell type ratios would be, however, still questionable. A more direct validation would be spatial transcriptomic techniques (e.g. 10x Visium). If none of these validations are possible all claims based on cell type abundances should be critically revisited or removed.*

Response

We agree on all points. Although we tried our best to avoid bias from the experimental procedure to reduce any batch effect, an impact on cell ratio from the individual differences and residual blood is clear (e.g., see responses 3.3 and 3.4, and elsewhere). Any claims about cell type abundances have been removed in our revision.

Regarding additional data types, *in situ* comparisons for lungfish remains challenging because of limited knowledge about its cell and tissue structure. We validated the expression and location of some marker genes in lung tissue using the new spatially-resolved transcriptomic technology STOmics and found several genes whose expression mirrored *in situ* hybridization (FISH) and single-cell marker expression data (**Figure R3.2**). However, the number of genes captured in our STOmics experiment was relatively low and will require some time to optimize for use with lungfish.

Figure R3.2. The cell type ratio of lungfish tissues using spatial transcriptomic test. The expression of marker genes from STOmics experiment.

Comment 3.6: 2. Staining

The validation of transcriptomic findings by *in situ* staining methods is commonly used for single-cell transcriptomics and I appreciate that the authors try to validate their findings given the fact that lungfishes are not a routine model organism. However, a greater magnification is needed to interpret the staining as it would be interesting to see whether the cell type specific genes identified are indeed expressed in separate cells in the tissue as suggested by the scRNA-seq results. With the resolution presented I don't see a clear difference between the staining for *npc2* and *arpc1b* even though the single-cell data suggests that the genes should be found in different cell types. I am not an expert in lung tissue architecture and hence I don't

know if these different populations are indeed in close proximity in the lung. Maybe a schematic could help to show that the cell types are expected to be found in the same part of the lungs. Generally, more explanations on how to read the images would be helpful.

Response

Firstly, we adjusted the magnification of *in situ* staining to show the cell outline clearer (**Figure R3.3A**). We found that macrophages (*arpc1b*) and alveolar cells (*npc2*) are distributed at the margin of the tissues, which is consistent with the results of H&E staining (**Figure R1.5**). We also verified the colocalization of these two genes on the same slide, showing that they are expressed in adjacent locations (**Figure R3.3C**). Secondly, based on the previous descriptions of the blood-gas barrier of the lung and the position of granulocytes in lungfish lung^{16 9}, we generated a diagram of the lung alveolar structure. The blood-gas barrier comprises endodermis, an interstitial layer, a basal lamina, and an epithelial layer. The granulocytes exist in the pulmonary capillaries and connective tissue underlying the epithelium (**Figure R3.3B**).

Figure R3.3. Validation of selected marker genes. (A) Fluorescence microscopy image of lungfish lung (left) and gill (right). Green, digoxigenin-labeled marker genes probes amplified using FITC-TSA; blue, DAPI. Yellow boxes indicate regions shown to the right of each tissue. (B) A schematic diagram of lungfish lung alveolar structure shows the major elements: alveolar cells, macrophages, epithelial cells, basal lamina, and a blood vessel. (C) Fluorescence microscopy image of lungfish lung. Green, digoxigenin-labeled marker genes probes of *arpc1b* (green) and *npc2* (red). Yellow boxes indicate regions shown to the right.

3. Cell type evolution

***Comment 3.7:** I think that the evolutionary perspective across different taxa ranging from fish to human is one of the most interesting aspects of the study presented. However, the methods used and the data presentation are not fully sound in my opinion.*

The interpretation of the cell type similarities is heavily dependent on the right assignment of cell types across the different data sets in the first place. The clusters were obtained unbiasedly and then afterwards assigned based on published marker genes for these data sets (human, mouse and salmon Fig. S5a-c) before similarity calls were made between different clusters. But there are explanations missing why the authors changed the strategy for the zebrafish data set or vice versa. What was the rationale for this and does it have an impact on the cluster similarity assignment?

Response

We appreciate this question. For the zebrafish data set, we directly used the cell type atlas from a recent publication ¹⁷ which contain all information for the cell type similarity analysis of this work. According to this reviewer's suggestion, we have also added the results of the cell atlas of zebrafish gill and swim bladder into the revised supplementary file (**Figure R3.4 A, B** below). Besides, the cell annotation was not provided directly in the supplementary data, but the primary cell count matrix, the markers genes of cell types, and bioinformatics pipeline of the human, mouse and salmon dataset are available. We reanalyzed the data using the similar methods and identified the cell types using the similar marker genes. In summary, all the cell assignment of published dataset is based on the similar method and cell markers from original papers, which are consistent with the published results.

Figure R3.4. The cell atlas of zebrafish tissues from ZCL dataset. (A) FFT-accelerated Interpolation-based t-SNE (Fit-SNE) of zebrafish gill cell types. Cells are color-coded by cluster cell type. Each dot represents a cell, and different colors are associated with specific cell types. (B) Fit-SNE of zebrafish swim bladder cell types. Annotated as in (A).

Comment 3.8: *As a proof for a correct cell type assignment and cluster alignment across the data sets the authors focus only on orthologs in the alveolar cells (Fig. 4b). But what is the pattern for the other cell types that are assigned across the different species? Are the other similarity assignments also meaningful on the gene level i.e. are there sets of transcription factors that allow a clear assignment to a certain cell lineage. In other words, I would like the authors to focus on the similarities rather than the differences which are mostly the focus of Figure 4. This would validate in parts the cluster assignments and is for the interpretation of evolutionary directions essential.*

Response

Thank you for the above suggestion. We presented the orthologs in the alveolar cells because it is one of the most important cell types in the lung. According to the reviewer's suggestion, we added more details of orthologs for similar cell types and reduced the part of the differences across species (**Supplementary_Data_13**).

In the gill comparison, many cell types show high conservation and share many meaningful genes among different species, especially the CD8+T cell, Epithelial_Ca, ionocyte cell, and macrophage. CD8+T cell in lungfish gills and immunocyte_tppp_high cell in zebrafish gills shared *ILF2*, a gene that regulates interleukin expression, and *PTPRC* (a regulatory factor of T-cell signal transduction).

The lungfish gill macrophage and salmon gill monocyte both highly expressed *SPI1*, which regulates the development of myeloid cells and B cells, and *ACP5*, expressed by macrophages and dendritic cells.

In the lung comparison, besides similarity at the cell type level, the macrophage, SMC, vascular endothelial cells, alveolar cells, and lymphoid cells share many meaningful genes among different species. The TFs *AEBP1* and *PRDM6* play a role in SMC differentiation and contraction and are expressed in the SMC of lungfish lung and the SMC and fibroblast of the mouse lung. Another example is the un_lymphoid of lungfish lung, NK cell of mouse, and CD8 T cells of human. They all expressed *IKZF1*, *MALT1*, *APBB1IP*, and other immune-related genes.

In summary, the cell similarity and associated gene expression both verify the conservation of the homologous organs across species. (Lines 294-372)

Figure R3.5. Cell type and features assignment across species. (A) Violin plots of orthologous genes with conserved expression in different cell types across species.

Comment 3.9: *Legend and method information are not available in a straightforward way. It is not clear to me what the grey lines in Fig. 4a are telling and based on which criteria they are drawn. Generally, a brief introduction of the method used to compare across species would be helpful and is currently missing in the text. The method was originally established for comparing 3 cnidarian whole body cell type atlases. A more detailed validation of the inferred cell type similarities by shared gene expression programs would not only improve the value of the comparisons but would also proof that the method applied not only works for a broad lineage assignment of cell types at the whole-body scale for the rather simple body plan of cnidarians (cnidocytes, neurons and gastrodermis) but also for the assignment of cell types on a more fine-grained level between lungfish and human lungs. I believe that critically evaluating the performance of the similarity assignment tool would be also interesting to other researchers working on scRNA-seq cross species comparisons.*

Response

Thank you. We have accordingly checked and rewritten the legend and method information in our revised manuscript.

Firstly, the Sankey plot in Fig. 4a shows the cell type similarity across lung tissues of different species. The first column represents the cell types from the zebrafish swim bladder, the second column represents the cell types from lungfish lungs, and the third column represent the cell types from the lung of mammals (human and mouse). Each light gray shadow means the covered cell types belong to the same broad cell types, including stromal, endothelial, epithelial, and immune cells. The grey line between specific cell types means the similarity of the top 5% highest Kullback–Leibler divergence (KDL) values. In short, the Sankey plot summarizes the circle plot information between the two species.

Secondly, our manuscript now has detailed descriptions of the cross-species method (**Lines 587-594**).

Finally, we agree with the reviewer's suggestion of using different tools to evaluate the performance of cross-species comparison. We employed frequently-used

methods to analyze our dataset. These included Seurat¹⁸, SAMap¹⁹, and CAME²⁰. The results are consistent between the different methods (**Figure R3.6**).

Comment 3.10: *In addition, the total number of orthologs identified is missing and should be added. If the number is large enough, can the datasets be integrated into one matrix to make the cell type assignment more coherent? Another idea in line with this could be to transfer the label from one data set to the other which is a function in the Seurat package which the authors are anyway using. In case the number of one-to-one orthologs is too small could the authors maybe use tools that are not based on this strict criterion such as SAMap to identify similar cell types? These are just some ideas that could also help to make the similarity calls between cell types more robust and reliable.*

Response

Thank you for the above constructive suggestions. The number of pairwise orthologs between lungfish and the other four species ranged from 8,485 to 11,674 (**Table R3.1**), which is large enough for downstream integration. We integrated these datasets into one matrix for cell type assignment using Seurat. The broad cell types were generally consistent (e.g., alveolar cells with ATI or ATII, endothelial cells, stromal cells, and some immune cell types) (**Figure R3.6**).

To confirm that the similarity calls between cell types were robust, we employed SAMap and the new tool CAME (cell type assignment and gene module extraction)²⁰, which showed general similar results (**Figure R3.6**), suggesting that our results are robust and reliable.

Table R3.1. Orthologs number between lungfish and other species.

Species	Orthologs number
Lungfish-Human	11,674
Lungfish-Mouse	11,191
Lungfish-Zebrafish	8,485

Lungfish-Salmon

9,347

Figure R3.6. Evaluation of different cross-species methods on the lungfish dataset. Each row represents the comparison of homologous organs from two species. Each column represents the result from a separate tool. KLD: Cell-cell similarities between lungfish and the other species. The top 5% highest Kullback–Leibler divergence (KDL) values are indicated as arches connecting cell types. Seurat: UMAP plots from integrated dataset of lungfish and other species. The cell types are labeled on the plot; the prefix of cell types is the name of the species. SAMap: Sankey plot of the cell-type mappings. Edges with alignment score <0.2 are omitted. CAME: The predicted cell-type probabilities for each cell (each column) in the lungfish lung (gill) scRNA-seq data. The gene expressions of the other species were taken as the reference. Each row indicates a cell type in the reference dataset.

4. Unsupported claims about immunity

Comment 3.11: *While mostly descriptive work, the authors are often pushing for speculative interpretations of the scRNA-seq results which are not supported in the current form of the manuscript. Examples are for instance found on lines 147-152, 197, 326, 355 or 372 and should be removed or moved in parts to the discussion section.*

To provide an example: “However, the human lung has developed additional functions, including an ostensibly stronger immune system”, is from an evolutionary perspective not appropriate and also not supported by simply finding more different cell types in the human lung scRNA-seq data.

Response

We have re-written these parts according to the Reviewer’s suggestion. We checked our manuscript thoroughly and removed speculative interpretations into the discussion section.

Minor points

Comment 3.12: • *Figure number assignments in the text should be double-checked, e.g.*

S4 to S5 line 510, Line 276/77 Fig. S3 assignment seems not correct

Response

We have reworded this sentence for clarity.

Comment 3.13: • *The color legend in Fig. 4c should be adjusted to correspond to the color panel in Fig. S5a for easier reading.*

Response

Thanks for your comment and we have redrawn the Fig. 4c to clearly show the cell types accordingly.

Comment 3.14: • *Increasing small font sizes in figures e.g. scale bar for in situ plots.*

Response

Thanks for your suggestion and we have readjusted the small font sizes of all the figures and hope to meet the request.

Comment 3.15: • *Reference 5 and 6 are duplicated*

Response

We are very sorry for our negligence of the reference arrangement. We have checked the reference again.

Comment 3.16: • *The title is a bit misleading as the single-cell study focuses only on lungs and gills and not the whole lungfish as “A single-cell transcriptome atlas of the West African lungfish ...” might suggest.*

Response

Thanks for your suggestion. We have revised the title to make it more appropriate with our manuscript. The edited title is “A single-cell transcriptome atlas of the West African lungfish lungs and gills highlights the respiratory evolution of a fish adapted to water and land”

Comment 3.17: • *More details in the figure captions are needed to understand the figures at the first glance.*

Response

Thank you so much for your careful check. We try to modified the figure captions to make it clear to read.

Comment 3.18: • *I would highly appreciate when the R codes used would be made publicly available (e.g. github). The methods section is in its current state often not helpful in understanding which exact commands were used when the authors refer to custom scripts.*

Response

We understand the reviewer's concern. We have collated the code we used and made it publicly available. The Github URL is <https://github.com/BGI->

Qingdao/Lungfish_scRNA_data_analysis. We have also rewritten the methods section.

Comment 3.19: • *I think that the in parts extended explanations about other lungfish species in the introduction are distracting from the actual species used in the study. I see potential to streamline the introduction when focusing more on the characteristics of the West African lungfish only.*

Response

Thank you for the above suggestions. We have revised the introduction and try to expressed the topic concentrating on the West African lungfish.

Comment 3.20: • *In contrast, the discussion has rather a summary character and could potentially be extended by moving in parts interpretations from the results section to this section.*

Response

We appreciate for your valuable suggestion. Considering the Reviewer's suggestion, we have expanded the content of discussion section and reduced redundancy description from the result part.

Reference

1. Cohen, M. *et al.* Lung Single-Cell Signaling Interaction Map Reveals Basophil Role in Macrophage Imprinting. *Cell* **175**, 1031-1044.e18 (2018).
2. Raredon, M.S.B. *et al.* Single-cell connectomic analysis of adult mammalian lungs. *Science Advances* **5**, eaaw3851 (2019).
3. Wang, Z. *et al.* Tumor-polarized GPX3⁺ AT2 lung epithelial cells promote premetastatic niche formation. *Proceedings of the National Academy of Sciences* **119**, e2201899119 (2022).
4. DePianto, D.J. *et al.* Molecular mapping of interstitial lung disease reveals a phenotypically distinct senescent basal epithelial cell population. *JCI Insight* **6**(2021).
5. Gaurav, R. *et al.* High-throughput analysis of lung immune cells in a combined murine model of agriculture dust-triggered airway inflammation with rheumatoid arthritis. *PLoS One* **16**, e0240707 (2021).
6. Deprez, M. *et al.* A Single-Cell Atlas of the Human Healthy Airways. *Am J*

- Respir Crit Care Med* **202**, 1636–1645 (2020).
7. Kim, E. *et al.* Maternal gut bacteria drive intestinal inflammation in offspring with neurodevelopmental disorders by altering the chromatin landscape of CD4+ T cells. *Immunity* **55**, 145–158.e7 (2022).
 8. Kim, E., Tran, M., Sun, Y. & Huh, J.R. Isolation and analyses of lamina propria lymphocytes from mouse intestines. *STAR Protocols* **3**, 101366 (2022).
 9. Jordan, H.E. & Speidel, C.C. Blood formation in the african lungfish, under normal conditions and under conditions of prolonged estivation and recovery. *Journal of Morphology* **51**, 319–371 (1931).
 10. Heimroth, R.D., Casadei, E. & Salinas, I. Effects of Experimental Terrestrialization on the Skin Mucus Proteome of African Lungfish (*Protopterus dolloi*). *Front Immunol* **9**, 1259 (2018).
 11. Heimroth, R.D. *et al.* The lungfish cocoon is a living tissue with antimicrobial functions. *Sci Adv* **7**, eabj0829 (2021).
 12. Tacchi, L., Larragoite, E.T., Muñoz, P., Amemiya, C.T. & Salinas, I. African Lungfish Reveal the Evolutionary Origins of Organized Mucosal Lymphoid Tissue in Vertebrates. *Curr Biol* **25**, 2417–24 (2015).
 13. Jaffer, T. & Ma, D. The emerging role of chemokine receptor CXCR2 in cancer progression. *Translational Cancer Research*, S616–S628 (2016).
 14. Fischer, U., Koppang, E.O. & Nakanishi, T. Teleost T and NK cell immunity. *Fish Shellfish Immunol* **35**, 197–206 (2013).
 15. Oikawa, T. *et al.* The role of Ets family transcription factor PU.1 in hematopoietic cell differentiation, proliferation and apoptosis. *Cell Death Differ* **6**, 599–608 (1999).
 16. Maina, J.N. & Maloiy, G.M.O. The Morphometry of the Lung of the African Lungfish (*Protopterus aethiopicus*): Its Structural–Functional Correlations. *Proceedings of the Royal Society of London. Series B, Biological Sciences* **224**, 399–420 (1985).
 17. Jiang, M. *et al.* Characterization of the Zebrafish Cell Landscape at Single-Cell Resolution. *Front Cell Dev Biol* **9**, 743421 (2021).
 18. Hao, Y. *et al.* Integrated analysis of multimodal single-cell data. *Cell* **184**, 3573–3587.e29 (2021).
 19. Tarashansky, A.J. *et al.* Mapping single-cell atlases throughout Metazoa unravels cell type evolution. *eLife* **10**, e66747 (2021).
 20. Liu, X., Shen, Q. & Zhang, S. Cross-species cell-type assignment from single-cell RNA-seq data by a heterogeneous graph neural network. *Genome Res* **33**, 96–111 (2023).

REVIEWER COMMENTS

Reviewer #1 (Remarks to the Author):

In this revised version of the manuscript, authors have addressed adequately some of the major concerns brought up by me and the other reviewers. However, new issues have emerged that have introduced inaccuracies to the paper. Some of these represent major concerns that need to be addressed. I would also like to mention that I would have loved to have a marked version of the manuscript with tracked or highlighted changes because it is a lot easier to review with the highlighted changes on.

1. The authors have of course concluded that presence of blood in their samples was a major confounding effect of their single cell results. However, as far as I can tell, only the control freshwater samples were perfused and blood removed to assess the impact of blood contamination which means that the terrestrialized samples still contain the contamination and therefore the comparisons are not accurate.
2. The histology (H&E) does not include panels from terrestrialized animals. This is critical to understand the changes in cell subsets identified in the scRNASeq
3. The in situs require some anatomical context. For instance: in the lungs delineate the alveolar space, cartilage in the gills etc
4. Figure R3.3 panel B: the schematic diagram of the lung is not correct at all. Authors can use their own H&E images or refer to Tacchi et al 2013 Developmental and Comparative Immunology to check the structure of a *Protopterus dolloi* lung.
5. The issues with assignments of immune cells is still there. For instance, myeloid cells contain macrophages and dendritic cells according to line 268, however, what markers were used to identify dendritic cells is unknown

Reviewer #2 (Remarks to the Author):

The revised version is very much improved and my points have all been addressed satisfactorily. I propose, as a service to the non-specialist reader, to include a sentence in the manuscript that the all currently possible approaches to identify the cell types may miss out on lungfish specific cell types (Comment 2.2)

Reviewer #3 (Remarks to the Author):

I am pleased to see that the authors addressed my concerns and suggestions. The manuscript is greatly improved. Particularly, I am glad to see that the cell type ratio inferences are removed from the manuscript. In addition, I appreciate that the methods section is reworked and that the codes are publicly available helping the community to better use the information provided in this original work. Furthermore, I acknowledge that the authors employed new methods which further validates and proves their findings.

Unfortunately, I am still not convinced by the interpretation of the immune system comparisons. The evaluation whether the immune system is stronger or weaker based on finding more or less different cell types of immune cells for species living in completely different habitats is not sound in my opinion. In addition, in its current form the text reads as if the immune system of mouse and lungfish are more similar than human and mouse which I strongly believe is not true given that human and mouse are both mammals.

"Taken together, our analysis suggests that the zebrafish swim bladder, West African lungfish and mouse lungs share immune cells, especially granulocyte which belong to the innate immunity system. However, the human lung has developed additional functions, including an ostensibly stronger immune system, which may indicate the elaborate and specialized cell type classification vary with the evolution."

I would like to see the above-mentioned summary as well as line 306 to be removed or strongly rephrased.

Otherwise, I would be happy to see the article published in its current form. There are only minor issues remaining:

- Sentence in line 242 needs to be reworked. And also the following sentence (line 243) is not fully clear to me as I am wondering how the lung can have more erythrocytes. To my knowledge there are no tissue resident red blood cells which this sentence suggests.
- 251 . should be ,

Point-to-point responses

Reviewer #1 (Remarks to the Author):

Comment 1.1: In this revised version of the manuscript, authors have addressed adequately some of the major concerns brought up by me and the other reviewers. However, new issues have emerged that have introduced inaccuracies to the paper. Some of these represent major concerns that need to be addressed. I would also like to mention that I would have loved to have a marked version of the manuscript with tracked or highlighted changes because it is a lot easier to review with the highlighted changes on.

Response

We thank the reviewer for the careful reading, helpful comments, and constructive suggestions, which have significantly improved the presentation of our manuscript. Our point-by-point responses are detailed below. The methods section has been thoroughly revised and edited by a native English speaker to improve grammar and spelling. We also provide a revised manuscript with tracked and highlighted changes.

Comment 1.2: 1. The authors have of course concluded that presence of blood in their samples was a major confounding effect of their single cell results. However, as far as I can tell, only the control freshwater samples were perfused and blood removed to assess the impact of blood contamination which means that the terrestrialized samples still contain the contamination and therefore the comparisons are not accurate.

Response

We thank the reviewer for the constructive suggestion. Per the reviewer's suggestion, we performed an additional terrestrialization experiment to assess the impact of blood contamination by adding more washing times. We added additional lung and gills data from a West African lungfish kept under terrestrialized conditions for 33 days, and our new results illustrate the reliability of biological and technical replications of the assigned cell types (**Figure R1.1, Extended Data Fig.1a, b**).

Then we focused on the content of immune cells which is the major effect of blood

contamination. For the number of immune cell comparison between the original and new protocol, we found the ratio of immune cells in lung samples to be similar (2.5% to 8.5%), while the ratio of immune cells in gill samples was notably reduced from 57.4% to 17.0% (Figure R1.1). The ratio of immune cells in lung samples is relatively stable, likely because of the thinness of the lungfish lung. We are very grateful for the reviewer's suggestion.

A

B

Figure R1.1. Effect on removal of blood on cell type assignment. Uniform Manifold Approximation and Projection (UMAP) of before (left) and after (right) additional washing steps to remove blood in the gill (A) and lung (B). Cells are color-coded by cluster cell type.

In conclusion, according to the reviewer's suggestion, we totally used two freshwater samples and a terrestrialized sample to assess the impact of blood contamination. We integrated the results of the two experiments and obtained 53,605 lung cells and 87,347 gill cells, which were assigned into 14 and 21 cell types, respectively. Compared to our initial manuscript submission, the cell type annotation greatly improved because there was a significant increase in the number of cells (29,640 to 53,605 in the lung, 19,385 to 87,347 in gill) (**Figure R1.2**).

Besides, we have revised all results and most figures and believe our data set and interpretations are indeed more robust. Marker genes and enrichment analysis were consistent, but, as expected, increasing the sample number to a minimum of three improved the data set. For example, we found that a marker of epithelial cells, *krt8*, now shows high expression levels in the lung epithelial cells. Similarly, we identify a downregulation of a solute carrier gene (*slc34a2*) in addition to *slc22a3* in terrestrialized lungfish lung alveolar cells. Our new data set also shows increased ligand-receptor (LR) pairs, from 33 to 35 in the lung and 29 to 51 in the gill. Finally, we detected more orthologs between the West African lungfish, mouse, and human alveolar epithelial cell types.

Figure R1.2. Sequencing of additional lungfish samples detects more cells and largely preserves the cell type assignments. Uniform Manifold Approximation and Projection of (A) gill and (B) lung cell types our initial submission (left panel) and a dataset including additional scRNA-seq samples. Cells are color-coded by cluster cell type.

***Comment 1.3:** 2. The histology (H&E) does not include panels from terrestrialized animals. This is critical to understand the changes in cell subsets identified in the scRNASeq*

Response

We agree with the reviewer that histology (H&E) is essential to understand the scRNA-seq cell clusters and now provide H&E images from terrestrialized and freshwater animals. We found that the basic structure and cell components of the lungs and gills are maintained in the two groups (**Figure R1.3**). This result agrees with our scRNA-seq data (**Extended Data Fig.2a,3a**). The macrophage and alveolar cells identified in the scRNA-seq data are observed in terrestrialized and freshwater lung H&E images (**Figure R1.3.A, B**). We also observed lymphoid nodes, vasculature, and macrophages in terrestrialized and freshwater gill H&E images. Those cell types are also found in our scRNA-seq data.

Figure R1.3. The H&E stain of African lungfish lungs and gill filament from freshwater sample.
 (A) The overall H&E staining image of terrestrialized lungfish lungs (left) and the Detail View Profile

of the red box (right). The black arrow represents alveolar epithelial cells; black asterisk means macrophage cells. **(B)** The overall H&E staining image of freshwater lungfish lungs (left) and the Detail View Profile of the red box (right). The labels are same as (A). **(C)** The overall H&E staining image of terrestrialized lungfish gill filament (left) and the Detail View Profile of the red box (right). The black asterisk represent macrophage, the red triangle means vessels and the red circle means lymphoid node. **(D)** The overall H&E staining image of freshwater lungfish lungs (left) and the Detail View Profile of the red box (right). The labels are same as (C).

Comment 1.4: 3. *The in situs require some anatomical context. For instance: in the lungs delineate the alveolar space, cartilage in the gills etc*

Response

This is a very helpful suggestion. We now include anatomical context on the in situs (**Figure R1.4**). We employed the lung structure of *Protopterus dolloi* (Nigerian lungfish) as a reference to label the lung anatomical structure ⁴. For the gill structure, we refer to *Protopterus aethiopicus* (marbled lungfish of Eastern and Central Africa) and zebrafish ^{3 5}.

Figure R1.4. The fluorescence microscopy image of lungfish lung and gills. (A) Fluorescence microscopy image of lungfish lung. Green, digoxigenin-labeled marker genes probes amplified using FITC-TSA; blue, DAPI. Fmw, fibromuscular wall; AS, air sacs. (B) Fluorescence microscopy image of lungfish gill. Cr, cartilage. (C, D, E) The revised figures of Fig.1f (C), g(D) and Extended Data Fig. 1f (E).1, fibromuscular wall; 2, air sacs. Fl, filament; Cr, cartilage; Wight arrows, lamella.

Comment 1.5: 4. Figure R3.3 panel B: the schematic diagram of the lung is not correct at all. Authors can use their own H&E images or refer to Tacchi et al 2013 *Developmental and Comparative Immunology* to check the structure of a *Protopterus dolloi* lung.

Response

We thank the reviewer for pointing out this error. According to the reviewer's suggestion, we consulted the histology (H&E) of *Protopterus dolloi* lung in the manuscript by Tacchi and colleagues. We have revised the schematic diagram, including pointing out the transverse section, and changed the scanning electron micrograph ⁶ to our own H&E image (Figure R1.5).

Figure R1.5. A schematic diagram of lungfish lung alveolar structure. (A) The previous schematic diagram of lungfish lung alveolar structure shows the transverse section (a). (B) A modified schematic diagram of lungfish lung alveolar structure shows the transverse section (a). The major elements of alveolar septa: alveolar cells, macrophages, capillaries and fibromuscular wall.

Comment 1.6: 5. The issues with assignments of immune cells is still there. For instance, myeloid cells contain macrophages and dendritic cells according to line 268, however, what markers were used to identify dendritic cells is unknown

Response

Thank you for raising this important point. As key components of the mucosal immune system ^{19,20}, we cannot rule out the existence of DCs in lungfish gills.

Previous studies have identified DC-like cells in the skin and gills of rainbow trout¹⁵, zebrafish¹⁶, and molly fish¹⁷. To date, no studies have reported the DCs in African lungfish gill. However, a morphological study of gills in the South American lungfish (*Lepidosiren paradoxa*) showed it harbors at least two types of leucocytes¹⁸. Based on the general naming rule of teleost fish gill DCs and our new integrated dataset (i.e., additional biological replicates), we believe we have identified dendritic-like cells in the lungfish gill. We combined cell markers from public databases and published research to define the DC-like cell type (**Figure R1.6**). In mammals, DCs and macrophage belong to the mononuclear phagocyte system (MPS)⁷. They have similar functions (such as phagocytosis and antigen presentation), while the DCs can modulate immune responses and stimulate T cells. The markers we used included *APOE*, *LGMN*, *VATI*, and *SEPP1* (**Figure R1.6.A**) -- four genes expressed in macrophages and DCs⁸. We also collected ostensible DC markers such *PTX3*⁹, *CYBB*¹⁰, *NFAM1*¹¹, *C10ORF54*¹², *KLRF2*¹³, and *TNFRSF21*¹⁴ (**Figure R1.6.B**).

Figure R1.6. The expression patterns in of lungfish macrophage and dendritic cell markers. (A) Bubble plot of typical granulocytes markers shows the expression pattern across cell types of lungfish gills. Circle size reflects the percentage of cells within a cell type which express the specific genes. The shades of color show average expression levels of specific gene. **(B)** Bubble plot of typical dendritic cell markers shows the expression pattern across cell types of lungfish gills. Annotated as in (A).

Reviewer #2 (Remarks to the Author):

Comment 2.1: The revised version is very much improved and my points have all been addressed satisfactorily. I propose, as a service to the non-specialist reader, to include a sentence in the manuscript that the all currently possible approaches to identify the cell types may miss out on lungfish specific cell types (Comment 2.2)

Response

We would like to thank the reviewer for the positive comments, and also the constructive suggestions. Following your suggestion, we have added the comments in the revised manuscript (**Lines 651-654**) and also as follows.

“It is important to note that current approaches to characterize cell types rely heavily on published markers from model animals. That is, particular West African lung fish cell types can be hard to classify or even identify (e.g., CTSG high cells in the gills with high levels of the antimicrobial protein cathepsin G in Fig. 1c).”

Reviewer #3 (Remarks to the Author):

Comment 3.1: I am pleased to see that the authors addressed my concerns and suggestions. The manuscript is greatly improved. Particularly, I am glad to see that the cell type ratio inferences are removed from the manuscript. In addition, I appreciate that the methods section is reworked and that the codes are publicly available helping the community to better use the information provided in this original work. Furthermore, I acknowledge that the authors employed new methods which further validates and proofs their findings.

Response

We thank the reviewer for the positive comments and for providing constructive suggestions. We think the revised manuscript has greatly improved based on your comments.

Comment 3.2: Unfortunately, I am still not convinced by the interpretation of the

immune system comparisons. The evaluation whether the immune system is stronger or weaker based on finding more or less different cell types of immune cells for species living in completely different habitats is not sound in my opinion. In addition, in its current form the text reads as if the immune system of mouse and lungfish are more similar than human and mouse which I strongly believe is not true given that human and mouse are both mammals.

"Taken together, our analysis suggests that the zebrafish swim bladder, West African lungfish and mouse lungs share immune cells, especially granulocyte which belong to the innate immunity system. However, the human lung has developed additional functions, including an ostensibly stronger immune system, which may indicate the elaborate and specialized cell type classification vary with the evolution."

I would like to see the above-mentioned summary as well as line 306 to be removed or strongly rephrased.

Response

We thank the reviewer's helpful comments. We agree with the reviewer's points that the statement on the immune system comparisons is inappropriate. We have revised the manuscript carefully according to your constructive suggestions as follows and removed the above-mentioned summary paragraph.

Line 306:

"However, many immune cell types in the human or mouse lung show no correlations with the West African lungfish (Extended Data Fig. 4e-f), likely reflecting the distinct immune cell repertoire of these terrestrial mammals."

Comment 2.3: *Otherwise, I would be happy to see the article published in its current form. There are only minor issues remaining:*

- *Sentence in line 242 needs to be reworked. And also the following sentence (line 243) is not fully clear to me as I am wondering how the lung can have more erythrocytes. To*

my knowledge there are no tissue resident red blood cells which this sentence suggests.

• 251 . should be ,

Response

We thank the reviewer for the careful review. According to your comments, we have revised the statement to make it more clearly in the manuscript and as follows. We removed the “more erythrocytes” in line 245 and the “However” in line 251.

Original Line 242-243 (New Line 355-359):

“The expression level of sema3a was too low to conclude an antagonist role in the West African lungfish. However, a previous study found that the lung of terrestrialized West African lungfish are better vascularized and expanded 15, hinting that SEMA3A outcompetes VEGFA vascularization effects under freshwater conditions despite its high gene expression.”

Line 251:

“After about a week of terrestrialization, the West African lungfish forms an.....”

References

1. Sturla, M., Paola, P., Carlo, G., Angela, M.M. & Maria, U.B. Effects of induced aestivation in *Protopterus annectens*: a histomorphological study. *J Exp Zool* **292**, 26-31 (2002).
2. Garofalo, F. *et al.* Signal molecule changes in the gills and lungs of the African lungfish *Protopterus annectens*, during the maintenance and arousal phases of aestivation. *Nitric Oxide* **44**, 71-80 (2015).
3. Laurent, P. *et al.* The vasculature of the gills in the aquatic and aestivating lungfish (*Protopterus aethiopicus*). *J Morphol* **156**, 173-208 (1978).
4. Tacchi, L., Misra, M. & Salinas, I. Anti-viral immune responses in a primitive lung: characterization and expression analysis of interferon-inducible immunoproteasome subunits LMP2, LMP7 and MECL-1 in a sarcopterygian fish, the Nigerian spotted lungfish (*Protopterus dolloi*). *Dev Comp Immunol* **41**, 657-65 (2013).
5. Dalum, A.S. *et al.* High-Resolution, 3D Imaging of the Zebrafish Gill-Associated Lymphoid Tissue (GIALT) Reveals a Novel Lymphoid Structure, the Amphibranchial Lymphoid Tissue. *Front Immunol* **12**, 769901 (2021).
6. Maina, J.N. The morphology of the lung of the African lungfish, *Protopterus aethiopicus*. *Cell and Tissue Research* (1987).
7. Chow, A., Brown, B.D. & Merad, M. Studying the mononuclear phagocyte system in the molecular age. *Nature Reviews Immunology* **11**, 788-798 (2011).

8. Franzén, O., Gan, L.-M. & Björkegren, J.L.M. PanglaoDB: a web server for exploration of mouse and human single-cell RNA sequencing data. *Database* **2019**(2019).
9. Garlanda, C., Bottazzi, B., Bastone, A. & Mantovani, A. PENTRAXINS AT THE CROSSROADS BETWEEN INNATE IMMUNITY, INFLAMMATION, MATRIX DEPOSITION, AND FEMALE FERTILITY. *Annual Review of Immunology* **23**, 337-366 (2005).
10. Keller, C.W. *et al.* CYBB/NOX2 in conventional DCs controls T cell encephalitogenicity during neuroinflammation. *Autophagy* **17**, 1244-1258 (2021).
11. Zhu, J. *et al.* Extracellular Vesicle-Derived circITGB1 Regulates Dendritic Cell Maturation and Cardiac Inflammation via miR-342-3p/NFAM1. *Oxid Med Cell Longev* **2022**, 8392313 (2022).
12. Lines, J.L. *et al.* VISTA is an immune checkpoint molecule for human T cells. *Cancer Res* **74**, 1924-32 (2014).
13. Alberts-Grill, N. *et al.* Dendritic Cell KLF2 Expression Regulates T Cell Activation and Proatherogenic Immune Responses. *J Immunol* **197**, 4651-4662 (2016).
14. Collin, M. & Bigley, V. Human dendritic cell subsets: an update. *Immunology* **154**, 3-20 (2018).
15. Soleto, I. *et al.* Identification of CD8 α (+) dendritic cells in rainbow trout (*Oncorhynchus mykiss*) intestine. *Fish Shellfish Immunol* **89**, 309-318 (2019).
16. Lugo-Villarino, G. *et al.* Identification of dendritic antigen-presenting cells in the zebrafish. *Proc Natl Acad Sci U S A* **107**, 15850-5 (2010).
17. Mokhtar, D.M. *et al.* Gills of Molly Fish: A Potential Role in Neuro-Immune Interaction. *Fishes* **8**, 195 (2023).
18. Wright, D.E. Morphology of the gill epithelium of the Lungfish, *Lepidosiren paradoxa*. *Cell and Tissue Research* **153**, 365-381 (1974).
19. Salinas, I., Fernández-Montero, Á., Ding, Y. & Sunyer, J.O. Mucosal immunoglobulins of teleost fish: A decade of advances. *Developmental & Comparative Immunology* **121**, 104079 (2021).
20. Mokhtar, D.M. *et al.* Main Components of Fish Immunity: An Overview of the Fish Immune System. *Fishes* **8**, 93 (2023).

REVIEWERS' COMMENTS

Reviewer #1 (Remarks to the Author):

The authors have addressed my concerns adequately. There are errors in the titles of figure legends.

For instance:

-Figure R1.3. The H&E stain of African lungfish lungs and gill filament from freshwater sample." The legend should say Morphological changes in the gill and lungs of freshwater and terrestrialized lungfish"

-Figure R1.4. The fluorescence microscopy image of lungfish lung and gills. The title of this figure should be "Confirmation of sc-RNA-Seq results by fluorescence in situ hybridization" or something like that.

I would double check the figure titles in all legends since the grammar and message are not up to the standards of the journal.

Reviewer #3 (Remarks to the Author):

The authors addressed all my concerns in their revised manuscript and I am happy to see that additional data has been added strengthening the comparative conclusions drawn in the manuscript.

I am happy to see the manuscript published in its current form.

Point-to-point responses

Reviewer #1 (Remarks to the Author):

Comment 1.1: The authors have addressed my concerns adequately. There are errors in the titles of figure legends.

For instance:

-"Figure R1.3. The H&E stain of African lungfish lungs and gill filament from freshwater sample." The legend should say Morphological changes in the gill and lungs of freshwater and terrestrialized lungfish"

-Figure R1.4. The fluorescence microscopy image of lungfish lung and gills. The title of this figure should be "Confirmation of sc-RNA-Seq results by fluorescence in situ hybridization" or something like that.

I would double check the figure titles in all legends since the grammar and message are not up to the standards of the journal.

Response

We thank the reviewer for the careful reading and also the constructive suggestions. According to the reviewer's suggestion, we have checked and rewritten the titles of the corresponding figure legends.

Figure R1.3. The morphological changes in the gill and lungs of freshwater and terrestrialized African lungfish.

Figure R1.4. Confirmation of scRNA-Seq cell type annotation results by fluorescence in situ hybridization of African lungfish lungs and gills.

Fig.1f. Confirmation of scRNA-Seq cell type annotation results by H&E staining and fluorescence microscopy image of African lungfish lung.

Fig.1g. Confirmation of scRNA-Seq cell type annotation results by H&E staining and fluorescence microscopy image of African lungfish gill.

Besides, we have carefully checked the figure titles of all legends, and mainly revised figure titles of Fig.1b-e, Fig.2-b, c, Fig.3a, Fig.4a and Fig.5a, b, d in the manuscript.

Reviewer #3 (Remarks to the Author):

Comment 3.1: *The authors addressed all my concerns in their revised manuscript and I am happy to see that additional data has been added strengthening the comparative conclusions drawn in the manuscript.*

I am happy to see the manuscript published in its current form.

Response

We thank the reviewer for the positive comments and all the constructive suggestions.

We think the manuscript has been greatly improved based on your suggestions.